



## Polar semi-volatile organic compounds in biomass burning emissions and their chemical transformations during aging in an oxidation flow reactor

Deep Sengupta,[1] Vera Samburova,[1] Chiranjivi Bhattarai,[1] Adam C. Watts,[1] Hans Moosmüller,[1] Andrey Y. Khlystov[1]

[1]Desert Research Institute, 2215 Raggio Parkway, Reno, NV 89512, USA

*Correspondence to*: vera.samburova@dri.edu

## Abstract

Semi-volatile organic compounds (SVOCs) emitted from open biomass-burning (BB) can contribute to chemical and physical properties of atmospheric aerosols and also may cause adverse health effects. The polar fraction of SVOCs constitutes a significant part of BB organic aerosols, and thus it is important to characterize the chemical composition and reactivity of this fraction. In this study, globally and regionally important representative fuels (Alaskan peat, Moscow peat, Pskov peat, Eucalyptus, Malaysian peat, and Malaysian agricultural peat) were burned under controlled conditions using the combustion chamber facility at the Desert Research Institute (DRI). Gas- and particulate-phase biomass-burning emissions were aged in an oxidation flow reactor (OFR) to mimic 5–7 days of atmospheric aging. Fresh and OFR-aged biomass-burning aerosols were collected on Teflon impregnated glass fiber filters (TIGF) in tandem with XAD resin media for organic compound (OC) speciation. The polar fraction extracted with dichloromethane and acetone was analyzed with gas chromatography mass spectrometry (GC-MS) for 84 polar organic compounds—including mono and dicarboxylic acids, methoxylated phenols, aromatic acids, anhydrosugars, resin acids, and sterols. For all these compounds, fuel-based emission factors (EFs) were calculated for fresh and OFR-aged samples. The carbon mass of the quantified polar compounds was found to constitute 5% to 7% of the total OC mass. High abundance of methoxyphenols (239 mg kg$^{-1}$ for Pskov peat; 22.6% of total GC-MS characterized mass) and resin acids (118 mg kg$^{-1}$ for Pskov peat; 14.5 % of total GC-MS characterized mass) was found in peat burning emissions (smoldering combustion). Concentration of some organic compounds (e.g., tetracosanoic acid) with molecular weight (MW) above 350 g mol$^{-1}$ decreased after the OFR aging, while abundances of low MW compounds (e.g., hexanoic acid) increased. This indicated a significant extent of fragmentation reactions in the OFR. Methoxyphenols decreased after OFR





aging, while a significant increase (3.7 to 8.6 times) in abundance of dicarboxylic acids emission
factors (EFs), especially maleic acid (10 to 60 times), was observed. EFs for fresh and ratios from
fresh-to-aged BB samples reported in this study can be used to perform source apportionment and
predict processes occurring during atmospheric transport.
**Keywords**. Biomass burning, organic aerosols, semi-volatile organic compounds (SVOCs), gas
chromatography, mass spectrometry, polar organic compounds, oxidation flow reactor



## 1    Introduction

Biomass burning (BB), including both wildfires and prescribed burns, is a major source of
carbonaceous aerosols in the atmosphere (Penner et al., 1991) and can contribute up to 75% of
total atmospheric aerosol mass loading (Andreae et al., 2001; Park et al., 2007). These
carbonaceous aerosols have significant impact on both regional and global radiative forcing
(Ramanathan and Carmichael, 2008). BB emissions also can cause adverse health effects (Arbex
et al., 2007; Regalado et al., 2006) because of the mutagenic property of particle-bound organic
compounds (Yang et al., 2010). Therefore, comprehensive, molecular-level characterization of BB
emissions is essential for understanding health effects. Such molecular characterization of BB
carbonaceous aerosols in the atmosphere, however, is challenging as these aerosols are composed
of tens of thousands of compounds (Goldstein and Galbally, 2007).

Current atmospheric chemistry models use a limited number of organic species because of the
complexity of atmospheric aerosol chemical composition and the lack of aerosol chemical
speciation data. Approximately 80% of BB organic mass emissions, especially aged emissions, are
not identified in such models (Bertrand et al., 2018; Jen et al., 2019), limiting the capabilities of
atmospheric organic aerosol modeling. Thus, improvement is needed in molecular-level speciation
of both fresh and aged BB emissions for more accurate model estimations.

Simulation of natural fires in a laboratory environment using a BB chamber is one way to
characterize the chemical composition of BB emissions (Yokelson et al., 2003). A number of
studies characterizing the molecular composition of combustion emissions from fuels that
represent different geographical regions have been completed: temperate conifers (Oros and
Simoneit, 2001a), deciduous trees (Oros and Simoneit, 2001b), grasses (Oros et al., 2006), and
peats (Samburova et al., 2016; Iinuma et al., 2007). Akagi et al. (2011) compiled fuel-based
emission factors (EFs) from different fuels from throughout the world, including the peatlands of
south Asia, and found that burning condition (flaming/smoldering) can influence the EFs of
individual compounds. These data have been used for modeling work in predicting ozone-forming
potential and other air quality impacts (Alvarado et al., 2015). Most of these source apportionment
studies, however, were focused on characterization of fresh emissions and emissions of either
particle-phase or gas-phase compounds.




Significant changes in organic aerosol composition during atmospheric transport have been
reported (Liu et al., 2017; Decker et al., 2019). These changes can impact local and regional air
quality. Also, the role of Siberian peat burning in haze formation in the Korean peninsula (Jung et
al., 2016) demonstrates the global impact of BB emissions and their atmospheric transport on
regional air quality. Some laboratory studies found an increase in organic aerosol (OA) mass after
photochemical aging (Ortega et al., 2013; Grieshop et al., 2009) while others observed a modest
decrease (Bhattarai et al., 2018). There is still limited data on evolution of chemical composition
of primary organic aerosols (POAs) during atmospheric aging. Some laboratory experiments
demonstrated degradation of levoglucosan (Hennigan et al., 2010; Kessler et al., 2010) and
oxidation of methoxyphenols in the gas phase (Yee et al., 2013) and aqueous phase (Net et al.,
2011). These studies have more mechanistic implications than quantifying gross change after
atmospheric oxidation. Recently, Fortenberry et al. (2018) characterized the chemical fingerprints
of aged biomass-burning aerosols (leaf and hardwood of white oak) by performing oxidation in a
potential aerosol mass oxidation flow reactor (PAM-OFR) and chemical analysis with a thermal
desorption aerosol gas chromatograph aerosol mass spectrometer (TAG-AMS). Bertrand et al.
(2018) analyzed 71 organic compounds in BB emissions, sampled from a smog chamber, with
high resolution time of flight mass spectrometry (HR-ToF-AMS). There is still a lack of
understanding, however, regarding (1) major organic compounds emitted from BB, (2) their roles
in atmospheric photochemical reactions, and (3) what compounds are responsible for light
absorption of fresh and aged BB emissions.

In this study, emissions from laboratory combustion of six globally important fuels (Alaskan peat,
Moscow peat, Pskov peat, Eucalyptus, Malaysian peat, and Malaysian agricultural peat) were
quantitatively analyzed for more than 250 individual organic species, and analyses of 84 polar
organic species is presented in this paper. BB emissions generated in a combustion chamber were
run though the OFR, mimicking approximately 5 to 7 days of atmospheric oxidation (Bhattarai et
al., 2018b), and the OFR output was analyzed to characterize aged BB emissions. BB emissions
were collected on filter and XAD media to identify distribution of organic species between the gas
and particle phases. For the polar fraction of collected organic compounds, we quantitatively
analyzed a total of 84 compounds (methoxyphenol derivatives, dicarboxylic acids,



monocarboxylic acids, aromatic acids, resin acids, and anhydrosugars). In the analyzed
anhydrosugars, we paid special attention to levoglucosan, a derivative from cellulose (Simoneit et
al., 1999), since levoglucosan has been widely used as a molecular tracer of BB emissions
(Bonvalot et al., 2016; Maenhaut et al., 2016). Methoxyphenols also have been used in source
apportionment studies (Schauer et al., 2001a; Schmidl et al., 2008b, 2008a). These source
apportionment studies, however, haven't combined such a wide range of different groups in a
single investigation. Here we provide a detailed targeted chemical analysis of both gas- and
particle-phase BB emissions from the combustion of individual biomass fuels from diverse
geographical locations for both fresh and aged emissions. EFs of gas- and particulate-phase
individual polar organic species are presented for six groups of compounds (methoxyphenols,
dicarboxylic acids, monocarboxylic acids, aromatic acids, anhydrosugars, and resin acids) and are
discussed in separate sections for fresh and OFR-aged BB samples. The fresh-to-aged ratio and
top contributing organic species also are discussed. The comparison between fresh and OFR-aged
BB emissions helps to understand the chemical evolution of BB plumes in the atmosphere and the
obtained data can be used in future source apportionment and atmospheric modeling studies.

## 2. Experiments

### 2.1 Fuel Description

We selected six globally and regionally important BB fuels: Alaskan peat, Moscow peat, Pskov
peat, Eucalyptus, Malaysian peat, and Malaysian agricultural peat. Five of these were peat fuels
selected from different geographical locations, representing smoldering combustion and one
(Eucalyptus) representing flaming combustion.
Peatland ecosystems, generally wetland or mesic ecosystems underlain by soils composed
primarily of partially-decomposed biomass, contain mostly organic carbon and more than 20%
mineral content, represent a vast terrestrial carbon pool, and are potentially vast sources of carbon
flux to the atmosphere during wildfires that consume peat (Harden et al., 2000). Peatlands in high-
latitude temperate and boreal regions are particularly vulnerable to increased fire-related carbon
emissions resulting from climatic warming and increases in fire season length, while peatlands in
low-latitude and tropical regions are threatened by factors such as deforestation for agriculture,



urbanization, and drainage (Turetsky et al., 2015). We collected Alaskan peat samples from the
upper 10 cm of soils within black spruce (*Picea mariana*) near crown forest (Chakrabarty et al.,
2016). High-latitude and Eurasia samples here are from *Sphagnum*- and cotton grass- (*Eriphorum*
spp.) dominated communities, collected from the Moscow (Odintsovo and Shatura districts) and
Pskov regions of Russia. These regions are representative of oligotrophic peat bogs found widely
across Siberia as well. Tropical peat in this study includes samples from two areas in Malaysian
Borneo. One set of samples is from a *Dipterocarp*-dominated lowland forest with largely intact
native land cover, while the second set is from a cleared agricultural area in the Kota Snamarahan
region.

We selected Eucalyptus because of its prevalence across Australia and its important contribution
to Australian wildland fires. In addition, economic losses and risk to life and property from fires
in eucalypt forests are magnified by their proximity to both fire-prone ecosystems and large urban
areas; often eucalypt-dominated stands form boundaries between these two land-use types. There
are nearly 900 species of the genera *Eucalyptus*, *Corymba*, and *Angophora*, which collectively
comprise woody plants known as eucalypts. Native to Australia, eucalypt-dominated forests cover
nearly 92 million ha (Hills, W.E.; Brown, 1978). In addition, the fast and hardy growth
characteristics of eucalypts have made them popular in warm ecoregions of Europe as well as
North and South America, where they readily escape cultivation and become established, dominant
community types near urban areas where they were originally introduced. Because of their high
oil content, rapid and dense growth, and vegetative structure, eucalypts are highly flammable and
contribute to high fire risk in areas where they occur (Goodrick and Stanturf, 2012).

**2.2 Reagents and Materials**
We obtained high-performance liquid chromatography (HPLC) grade methanol and hexane from
Fisher Scientific (Fair Lawn, NJ, USA) and used the following filters for sampling and further
chemical analyzes: pre-fired (900 °C for 4 h) 47 mm diameter quartz-fiber filter (2500 Pallflex
QAT-UP, Pall Life Sciences, Ann Arbor, MI, USA) for thermo-optical Elemental Carbon/Organic
carbon (EC/OC) analysis, Teflon® filters (2500 Pallflex QAT-UP, Pall Life Science, Ann Arbor,
MI, USA) for gravimetric particulate matter (PM) mass analysis, and Teflon-impregnated glass



fiber (TIGF) 47 mm diameter filters (Fiber FilmT60A20, Pall Life Sciences, Ann Abor, MI, USA)
for organic analysis. We purchased the following deuterated internal standards from Cambridge
isotope laboratories (Tewksbury, MA, USA)  and CDN isotopes (Pointe-Claire, Quebec, Canada):
hexanoic-d11 acid, benzoic-d5 acid, succinic-d4 acid, decanoic-d19 acid, adipic-d10 acid, suberic-
d12 acid, levoglucosan-13C6, homovanillic-2,2-d2 acid, myristic-d27 acid, heptadecanoicd33
acid, oleic-9,10-d2 acid, tetradecanedioic-d24 acid, and cholesterol-2,2,3,4,4,6-d6.


**2.3 Biomass Burning (BB) Experiments**

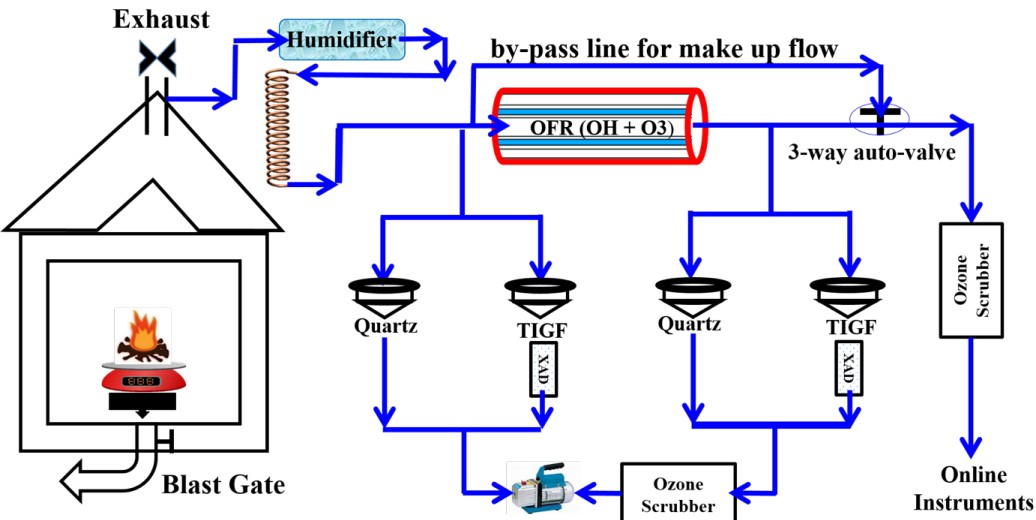


**Figure 1.** Desert Research Institute (DRI) biomass burning (BB) facility with oxidative flow
reactor (OFR) and flow setup.

BB experiments were conducted using DRI's BB facility for combustion of the selected fuels
under controlled conditions. A close replicate of this facility was described previously (Tian et al.,
2015), and a detailed description of the experimental setup was presented elsewhere (Bhattarai et
al., 2018; Sengupta et al., 2018).

We mixed laboratory-generated BB emissions with humidified zero air (Airgas Inc., Sparks, NV,
USA) using 4 m long spiral copper tubing (12.7 mm OD). Before it was mixed with the BB





emissions, the zero air was humidified by bubbling through Nano-pure water in a glass 500 mL
volume impinger. The flow rate was controlled with a mass flow controller (810C-CE-RFQ-1821,
Sierra Instruments, Monterey, CA, USA). An oxidation flow reactor (OFR) (Aerodyne Research
Inc., Billerica, MA, USA) was used to mimic approximately seven days of equivalent atmospheric
aging (Bhattarai et al., 2018a). The OFR consisted of an alodine-coated aluminum cylinder (46 cm
length and 22 cm diameter) with an internal volume of 13.3 L. Two sets of lamps emitted UV
radiation at wavelengths of 185 and 254 nm (Atlantic Ultraviolet Corporation, Hauppauge, NY,
USA) in the OFR to produce ozone and OH radicals (Li et al., 2015). UV irradiance in the OFR
was quantified using a photodiode detector with a wavelength range of 225 to 287 nm
(TOCON_C6; Sglux GmbH, Berlin, Germany). Ultra-high-purity nitrogen (Airgas Inc., Reno,
NV, USA) was used to purge the UV lamp compartments to prevent the lamps from overheating.
A probe that monitored relative humidity and temperature inside the OFR (from Aerodyne Inc.,
MA, USA) was mounted toward the outlet side of the OFR. A detailed characterization of the
OFR—such as particle loss, OH production rate, and time scales of various processes—can be
found in Bhattarai et al. (2018).

The duration of smoldering combustion experiments ranged from 69 to 255 min, whereas the
average duration of flaming combustion experiments was 50 min. During all experiments, both
fresh (directly from the chamber) and aged (oxidized in the OFR) emissions were continuously
collected on a TIGF filter (for particle phase) followed by an XAD cartridge (for gas phase) for
detailed chemical speciation. We used several online instruments to characterize gas- and particle-
phase pollutants (see Fig. 1). Simultaneous collection of samples for thermal optical carbon
analysis on quartz fiber filters (Pall-Gelman, 47 mm diameter, pre-heated) was conducted, but only
for Eucalyptus and Malaysian peat. The online instruments alternated every 10 min between
sampling fresh and aged emissions using a computer-controlled valve system.

We employed a bypass flow to keep the flow from the BB chamber and through the OFR constant
when online instruments switched between sampling fresh and aged emissions. To protect online
instruments from high ozone concentrations produced in the OFR, ozone scrubbers were installed
in front of the instruments' inlets. The ozone scrubbers were loaded with charcoal followed by
Carulite 200 catalyst (Carus Corp., Peru, IL, USA). There were no ozone scrubbers before the



filter-XAD set up, which could cause further oxidation of organic compounds on filter surfaces
during sampling. The reaction rates between organics and ozone, however, are orders of magnitude
lower than OH oxidation reactions (Finlayson-Pitts and Pitts Jr, 1999). Therefore, we assumed that
reactions with OH radicals were primarily responsible for changes in organic compounds
associated with fresh gas and particulate emissions.

## 220 2.4. Organic and Elemental Carbon (OC/EC) Analysis

Emissions from the combustion of two fuels (Eucalyptus and Malaysian peat) were sampled with
quartz-fiber filters, collected simultaneously with TIGF filters, for both fresh and aged BB aerosols
(Supplementary Material, Fig. S1). Punches (area = 1.5 cm$^2$) from these quartz filters were
analyzed with a thermal-optical carbon analyzer (Atmoslytic Inc., Calabasas, CA, USA) following
the IMPROVE protocol (Chow et al., 1993, 2004) for total organic carbon (OC$_{Total}$) and elemental
carbon (EC) mass.

## 228 2.5 Analytical Methodology for GC-MS

We extracted filter and XAD samples for GC/MS analysis (SI Table S1) yielding concentrations
of 84 polar organic compounds. In addition, levoglucosan concentrations were determined using
ion chromatography coupled with a pulsed amperometric detector (IC-PAD). Prior to the
extraction, sampled TIGF filters and XAD-resin cartridges were spiked with deuterated internal
standards (see "Reagents and Materials" section). The TIGF filters and XAD cartridges were
extracted separately with an accelerated solvent extractor (Dionex ASE-300, Sunnyvale, CA,
USA) at the following conditions: 80° C temperature, 250 mL extraction volume, and subsequent
extraction with dichloromethane and acetone. The XAD and filters were treated separately to
evaluate the speciation of gas- and particle-phase semi-volatile polar compounds. The extracts
were concentrated with a rotary evaporator (Buchi-R124, Switzerland), filtered using 0.2 μm pore
size syringe filters (Thermo Scientific, Redwood, TN, USA), and pre-concentrated with nitrogen
to a volume of 4 mL. Then we split the extracts into two fractions. One fraction was transferred to
2.0 mL volume deactivated glass maximum recovery vials (Waters Corporation, Milford, MA,
USA), pre-concentrated to 50 μL volume under ultra-high-purity nitrogen (Airgas, Reno, NV,
USA), and derivatized with N,O-bis-(trimethylsilyl) trifluoroacetamide (BSTFA with 1% of




trimethylchlorosilane; Thermo-Scientific, Bellefonte, PA, USA) and pyridine as described
elsewhere (Rinehart et al., 2006). Derivatized samples were analyzed by electron impact ionization
using a Varian CP-3400 gas chromatograph with a CP-8400 auto-sampler and interfaced to a
Varian 4000 ion trap mass spectrometer (Varian Inc. Palo Alto, CA, USA). The second fraction of
non-derivatized extracts was kept for further analysis of non-polar organic species (e.g., alkanes
and PAHs), and those results will be presented in future publications.

**2.6. Levoglucosan Analysis**
Portions of quartz filters collected for OC/EC analysis also were used for quantitative analysis of
levoglucosan concentration with IC-PAD. Prior to the analysis, quartz filters were extracted with
15 ml of deionized water (18.2 M$\Omega$), sonicated for one hour, and refrigerated overnight. The
column temperature for IC was 25° C. Analytes along with a mixture of two eluents (48%
hydroxide solution and 52% deionized water) were passed through the IC column with a 0.4 ml
min$^{-1}$ eluent flow rate and detected using an electrochemical detector. See Chow and Watson
(2017) for details**.**



**3. Results and Discussion**

**3.1. Gas- and Particulate-Phase Emission Factors**
Organic compounds (84 in total) in fresh emissions identified and quantified in this study were
assigned to six major groups (Table S1): methoxyphenol derivatives, dicarboxylic acids, mono-
carboxylic acids, aromatic acids, resin acids, and levoglucosan. First, we report individual
emission factors (EF) belonging to a particular group calculated by summation of gas- and particle-
phase EFs of individual compounds. Relative abundance of these compounds are reported next
followed by a comparison of the contributions of each group (EF$_{group}$) among fuels and a
comparison with previously reported results.
*3.1.1 Methoxyphenol Derivatives*



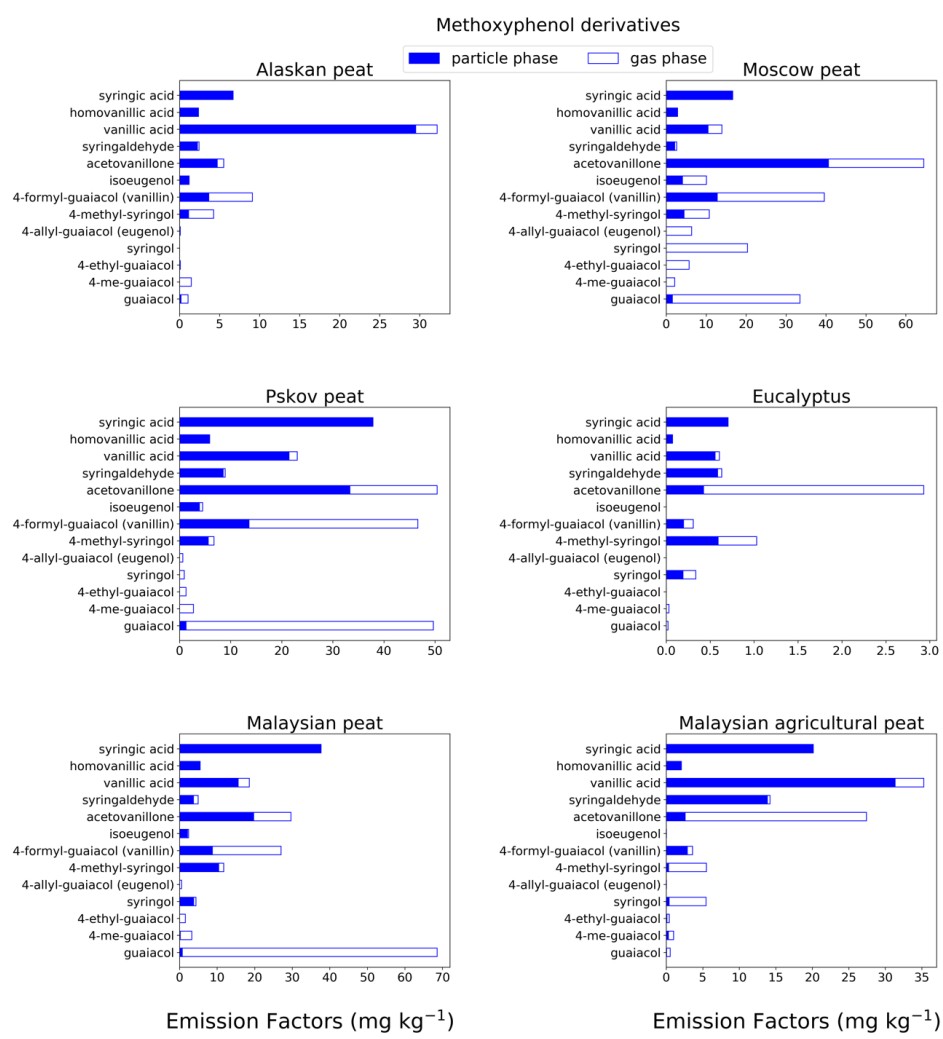


Figure 2a. EFs for methoxyphenols in both particulate phase (solid bars, filter samples) and gas-phase (open bars, XAD samples) from fresh biomass burning emissions for six different fuel types. We did not burn fuels in replicates, and standard deviations (SD) were calculated based on replicate analysis of emissions from similar fuels (with identical experimental conditions) during our previous combustion campaigns (Yatavelli et al., 2017a) where SD ranged between 9.7 and 22% for methoxyphenol derivatives.






Methoxyphenols are key compounds in BB smoke since they constitute from 20 to 40% of total
identified organic aerosol mass (Hawthorne et al., 1989; Yee et al., 2013). For this reason, these
compounds are considered potential markers for wood combustion (Schauer et al., 2001b) and
have been used as probable biomarkers to determine human exposure to BB emissions (Simpson
and Naeher, 2010; Dills et al., 2006). Our analysis of 13 methoxyphenols (Fig. 2a, Table S1)
showed that guaiacol (MW = 124±12 g mol$^{-1}$) was the major contributor to EFs of the measured
methoxyphenols in Moscow peat (33.5±3.3 mg kg$^{-1}$), Pskov peat (49.7±4.8 mg kg$^{-1}$), and
Malaysian peat (68.6±6.7 mg kg$^{-1}$). Syringol, another methoxyphenol commonly found in BB
emissions (Schauer et al., 2001a), had the highest EF for Moscow Peat fresh emissions (20.4±2.7
mg kg$^{-1}$), while for the other fuels, the EF was much lower (0–5.5 mg kg$^{-1}$). EFs for syringic acids
(MW = 198 g mol$^{-1}$) were in the range of 0.06–37.9 mg kg$^{-1}$ for all fresh emissions. Syringols are
generally not formed during pyrolysis of coniferous lignin, but during pyrolysis of deciduous
lignin, where both guaiacols and syringols are formed (Mazzoleni et al., 2007). Presence of both
guaiacol and syringol moieties in fresh emissions indicates that the part of the plant material that
was responsible for peat formation was probably from deciduous trees, and this signature of
deciduous trees from peat burning emission is irrespective of geographical origin of those peats
(also shown by Schauer et al., 2001a). Acetovanillone, vanillin, and vanillic acid also were
observed in fresh emissions with high abundance (5–50 mg kg$^{-1}$). For example, vanillin is an
abundant methoxyphenol in the fresh emissions from Pskov peat (46.7±5.4 mg kg$^{-1}$) which
contributed 4.4% of the total mass of the 84 analyzed compounds.

Low molecular weight methoxyphenols (e.g., guaiacol) are expected to be found in the gas phase
(Yatavelli et al., 2017b), in close agreement with our results. For example, guaiacol and substituted
guaiacols were mostly present in gas phase (82–100%) for emissions from the combustion of
different fuels (Fig. 2a, Table S1). With the addition of more oxygenated functional groups to a
molecule, and thus with molecular weight increase, the equilibrium gas-particle partitioning of the
compound tends to shift toward the particulate phase, which also was confirmed by our results
(e.g., for acetovanillone, a keto form of lignin derivative, from Malaysian peat combustion, 33.5%
of its mass was found in gas phase; for more oxygenated syringic acid, 99% of its mass was found
in particulate phase emissions from the same fuel).






The highest methoxyphenol $EF_{group}$ from combustion of all fuels was observed in the fresh Pskov
peat (Fig. 3a) emissions (239±11 mg kg $^{-1}$). For Moscow peat, which was sampled close to the
geographical region of Pskov peat, the $EF_{group}$ of methoxyphenols was 229±10 mg kg $^{-1}$ (Fig. 3a),
very similar to that for Pskov peat. The methoxyphenol $EF_{group}$ for peat samples were in the range
of 66 to 239 mg kg $^{-1}$ (Fig. 3a) for our 13 analyzed compounds. A previous study analyzed for 30
different compounds (Schauer et al., 2001a) and consequently found a larger value $EF_{group}$ of up
to 1330 mg kg $^{-1}$, at least partially a result of the larger number of compounds analyzed. Formation
of methoxyphenols during biomass combustion is mainly because of pyrolysis of lignin (e.g.,
Simonelt et al., 1993). Lignin, an essential biopolymer of wood tissue, is primarily derived from
three aromatic alcohols: p-coumaryl, coniferyl, and sinapyl alcohols (Hedges and Ertel, 1982).
Lignins of hardwoods (angiosperms) are enriched with products from sinapyl alcohol; softwoods
(gymnosperms) instead have a high proportion of products from coniferyl alcohol with a minor
contribution from sinapyl alcohol; grasses have mainly products from p-coumaryl alcohol. The
relative proportions of these bio-monomers vary considerably among the major plant classes
(Sarkanen and Ludwig, 1971), reflected in our total emission factors estimate for 13
methoxyphenols.














**_3.1.2 Dicarboxylic Acids_**

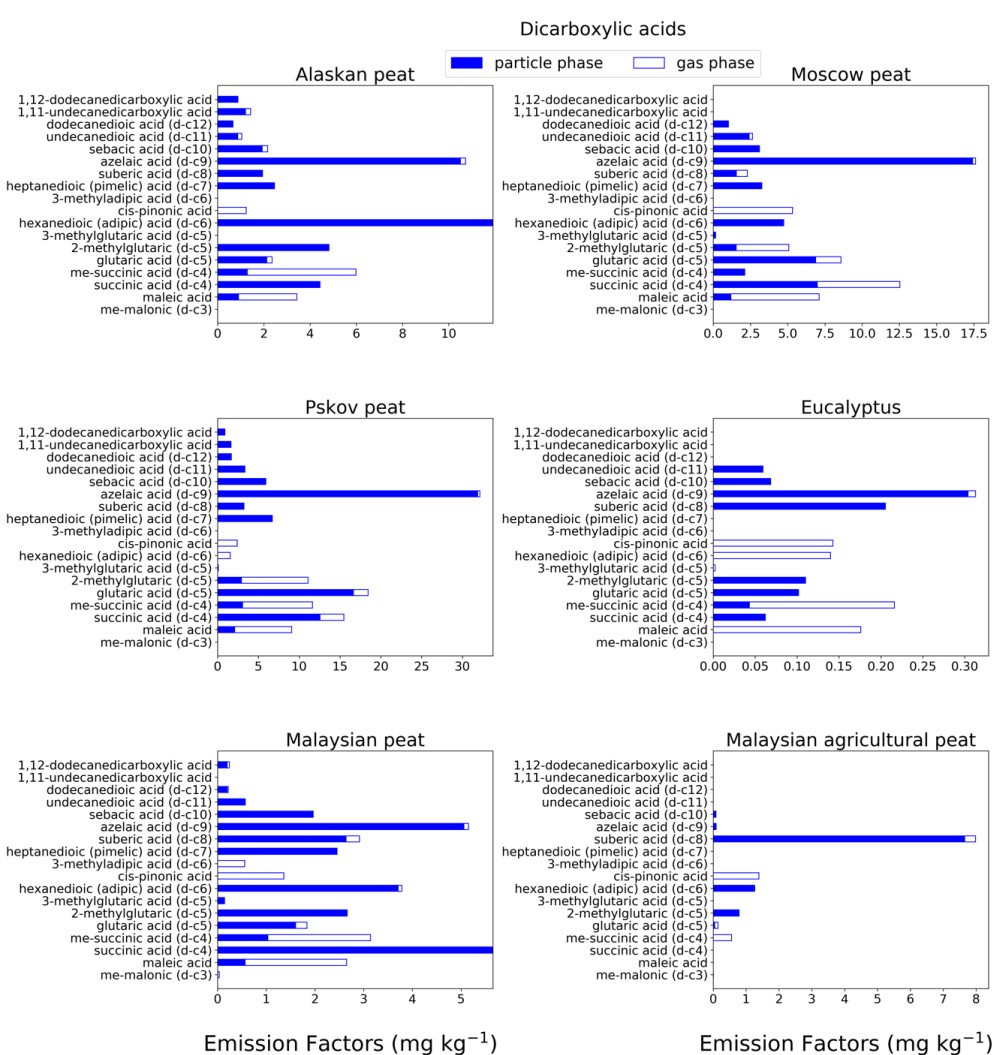

Figure 2b. EFs for dicarboxylic acids in both particulate phase (solid bars, filter samples) and gas-phase (open bars, XAD samples) from fresh biomass-burning emissions for six different fuel types. We did not burn fuels in replicates, and standard deviations (SD) were calculated based on replicate analysis of emissions from similar fuels (with identical experimental conditions) during our previous combustion campaigns (Yatavelli et al., 2017a) where SD ranged between 10 and 17% for dicarboxylic acids.






Dicarboxylic acids play a significant role in the atmospheric organic aerosols budget (Samburova
et al., 2013; Yatavelli et al., 2017b) via secondary organic aerosol formation that either changes
radiative forcing directly, or indirectly by acting as cloud condensation nuclei (Kawamura and
Bikkina, 2016). The $EF_{group}$ for dicarboxylic acids (Fig. 3b) varied among the fuels with the highest
EF for fresh Pskov peat samples (123±10 mg kg $^{-1}$) and with the lowest for Eucalyptus (1.5±0.1
mg kg$^{-1}$). This range in EFs can be attributed to difference in fuel type and burning conditions
(smoldering vs. flaming). We also observed, however, a difference in the $EF_{group}$ of dicarboxylic
acid between two tropical peats from the same geographical area (Malaysian peat: EF=35.33±2.9
mg kg$^{-1}$ and Malaysian agricultural peat: 12.29 ±1.02 mg kg$^{-1}$). The highest EF for individual
dicarboxylic acids was observed for azelaic acid. For example, for Pskov peat the EF was 32.1±4.1
mg kg$^{-1}$; and for Moscow peat it, was 17.6±2.6 mg kg$^{-1}$. Azelaic acids were mostly found in the
particulate phase (Fig. 2b, Table S1) and their relative abundances in the gas phase varied between
0.77% (for Pskov peat) and 2.85% (for Eucalyptus. Maleic acid was mostly found in the gas phase
(73%–83%), since it is a lower MW compound (MW=116.0 g mol$^{-1}$) compared to azelaic
(MW=188.22 g mol$^{-1}$) and adipic (146.14 g mol$^{-1}$) acids. Succinic and methyl-succinic acids are
found in both gas and particulate phases (Table S1), and their abundance in the particulate phase
was 19–59% and 53–100%, respectively. For Malaysian peat BB emissions, succinic acid was
present only in the particulate phase. A distinguishable increase in dicarboxylic acid mass
concentrations was observed for ambient aerosols followed by a biomass burning event (Cao et
al., 2017) compared to normal ambient concentrations. The formation of saturated dicarboxylic
acids (e.g., succinic acid) and unsaturated dicarboxylic acids (e.g., maleic acid) also was reported
for ambient aerosols collected near a biomass-burning event (Graham et al., 2002; Kundu et al.,
2010; Zhu et al., 2018) and in ice core records historically affected by biomass burning (Müller-
Tautges et al., 2016).



*3.1.3 Monocarboxylic Acids*

Figure 2c. EFs for monocarboxylic acids in both particulate phase (solid bars, filter samples) and

gas-phase (open bars, XAD samples) from fresh biomass burning emissions for six different fuel

types. We did not burn fuels in replicates and standard deviations (SD) were calculated based on



replicate analysis of emissions from similar fuels (with identical experimental conditions) during
our previous combustion campaigns (Yatavelli et al., 2017a) where SD ranged between 9.4 and
12% for monocarboxylic acids.

Monocarboxylic acids can constitute up to 30–40% of total identified organic aerosol mass from
BB emissions (Oros et al., 2006). In our study, we characterized the range from $C_6$–$C_{24}$, where
some unsaturated monocarboxylic acids (e.g., oleic acid) also are included. For Alaskan and
Malaysian peat fresh emissions (Fig. 2c), the highest EF (gas + particle) among all analysed
monocarboxylic acids was for hexadecanoic acid ($C_{16}$) with EFs of 55.7±6.6 mg kg$^{-1}$ and 51.8±6.2
mg kg$^{-1}$, respectively. The dominance of hexadecanoic acid among other monocarboxylic acids in
combustion emissions also was observed in ambient measurements during biomass-burning events
in southeast Asia (Fang et al., 1999). For Moscow (41.5±6.5 mg kg$^{-1}$) and Pskov (49.8±7.8 mg kg$^{-1}$
$^1$) peats, tetradecanoic acid ($C_{14}$) had the highest EFs in fresh samples (Fig. 2c). For Eucalyptus
and Malaysian agricultural peat fresh samples, the largest contributor to monocarboxylic acids was
tetracosanoic acid ($C_{24}$) (Fig. 2c) with EFs of 3.81±0.5 mg kg$^{-1}$ and 42.0±5.1 mg kg$^{-1}$, respectively.
As we expected, low molecular weight monocarboxylic acids like hexanoic acid (MW=116 g mol$^{-1}$
$^1$) was mostly present in the gas phase, and the gas phase mass fraction varied between 72% (for
Malaysian agricultural peat) and 98% (for Moscow peat). Similar trends were observed for other
low molecular weight monocarboxylic acids. For example, the relative abundance of octanoic acid
($C_8$) in the gas phase was 93.4% for Alaskan peat. High molecular weight monocarboxylic acids
($C_{16}$>) abundance in the gas phase was < 2% for all analyzed fuels.

Carbon preference index (CPI) is generally used for source apportionment of organic aerosols
(Fang et al., 1999). We also computed the carbon preference index for monocarboxylic acids from
all analyzed fuel combustion emissions by taking even carbon number over odd carbon number
ratio on EFs of monocarboxylic acids ranges from $C_6$ to $C_{24}$ (Fig. S1). For fresh emission samples,
the CPI values ranged from 1.28 (for Moscow peat) to 4.53 (for eucalyptus). The CPI values are
higher for fresh emissions of tropical peats (for example, 2.78 for Malaysian peat) than for
emissions from peats from high latitudes (for example, 1.74 for Pskov peat). An average CPI index
of 3.7 for monocarboxylic acids was reported for combustion emissions from sedimentary bogs



(Freimuth et al., 2019), in the range of our reported values (CPI: 1.3–4.5).

The sums of the EFs for 25 monocarboxylic acids are shown in Fig. 3c. The $EF_{group}$ was in the
range of 5 and 515 mg kg$^{-1}$ for all fuels. This range is comparable to the $EF_{group}$ reported previously
for grass (tundra, cotton, Pampas and ryegrass) combustion (32–250 mg kg$^{-1}$) (Oros et al., 2006).
Overall, the trend of low EFs associated with flaming combustion of Eucalyptus (16±0.7 mg kg$^{-1}$)
compared to smoldering peat combustion is also evident for this compound category. Combustion
of peat fuels from tropical regions (e.g., Malaysian agricultural peat) resulted in monocarboxylic
acids EFs of 212±9.6 mg kg$^{-1}$ compared to higher EFs from Alaskan peat combustion (505±23 mg
kg$^{-1}$). The origin of monocarboxylic acid is mostly plant wax and oils (Simoneit, 2002). The
relative proportion of plant wax and oils can vary widely among vegetation taxa and also their
concentrations in peat depend on biogeochemical processes involved in peat formation. The
differences in relative abundance of waxes and plant oils in living vegetation and the differences
between biogeochemical processes involved in peat formation for arctic and tropical regions may
be responsible for diverse EFs for monocarboxylic acids.
















*3.1.4 Aromatic Acids*

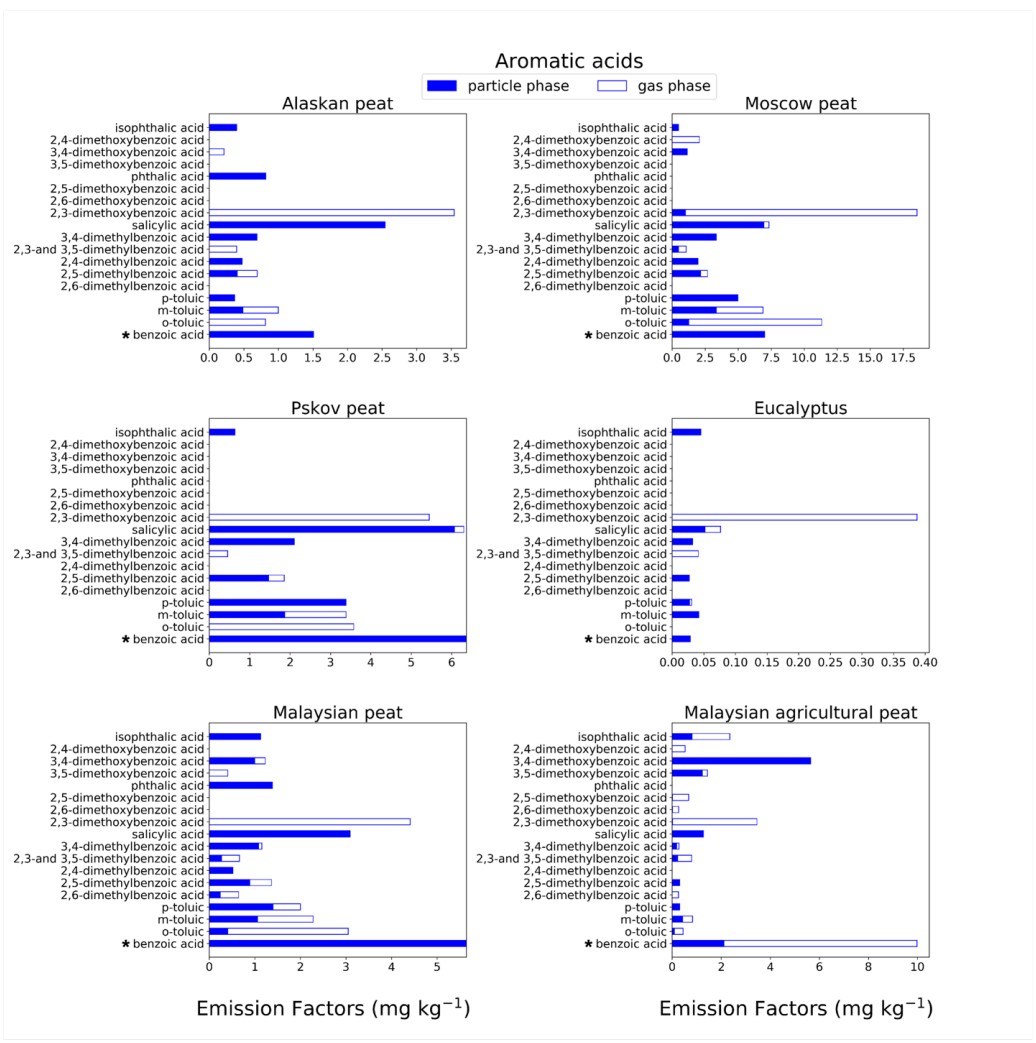


Figure 2d. EFs for aromatic acids in both particulate phase (solid bars, filter samples) and gas-
phase (open bars, XAD samples) from fresh biomass burning emissions for six different fuel types.
We did not burn fuels in replicates, and SD were calculated based on replicate analysis of
emissions from similar fuels (with identical experimental conditions) during our previous
combustion campaigns (Yatavelli et al., 2017a) where SD ranged between 9.5 and 15% for
aromatic acids; *Benzoic acid was found in high concentrations in the XAD blanks that introduced
a substantial uncertainty to quantification of this compound.






Aromatic acids from BB emissions can contribute up to 20–35 % of total identified organic mass
(Wan et al., 2019). In our study, the aromatic acids (e.g., p-hydroxy benzoic acid), excluded
methoxyphenol derivatives and resin acids. For most of the fuels, LMW aromatic acids (MW <150
g mol$^{-1}$) (e.g., benzoic acid, o-/m-/p-toluic acids) contributed more (almost 40% of total aromatic
acid emissions) toward total fresh emissions, compared to HMW aromatic acids (MW >150 g mol$^{-1}$
$^{1}$). For example, the benzoic acid EF for Malaysian agricultural peat fresh emission (Fig. 2d) is
9.98±1 mg kg$^{-1}$, and the EF for the same benzoic acid is 6.36±0.6 mg kg$^{-1}$ for Pskov peat (Fig. 2d).
Although, o-toluic and p-toluic acids are found in gas phase with 50%–100% abundance in
Alaskan and Moscow peat, benzoic acid is found only in particulate phase. Benzoic acid was found
in high concentrations in the XAD blanks that introduced a substantial uncertainty to quantification
of this compound . One of the most abundant aromatic acids in fresh peat emissions was 2,3-
dimethoxy benzoic acid. For example in Moscow peat, the EF was 18.6±4.7 mg kg$^{-1}$. The acid was
mostly found in gas phase (91%–100%) for all fuels (Fig. 2d, Table S1). 2,3-dimethoxy benzoic
acid is potentially derived from combustion of lignin moieties of biomass, and the emission of this
compound is more than an order of magnitude lower in emissions from flaming combustion
samples (EF=0.38 ±0.09 mg kg$^{-1}$) than in emissions from smoldering combustion emissions
(EF=5.44±1.36 mg kg$^{-1}$). Emissions of 3,4 dimethoxy benzoic acid were only observed for peats
from tropical regions (EF=5.64±0.8 mg kg$^{-1}$) with 80–100% abundance in particulate phase. This
compound can be used for source apportionment of aerosols emitted from burning of tropical peat
and also can potentially help to distinguish between emissions from tropical and high latitude
peatland fires.

The EF$_{group}$ for aromatic acids in fresh combustion emissions from eucalyptus fuel is extremely
low (0.71±0.05 mg kg$^{-1}$) compared to that for peat fuels (13–69 mg kg$^{-1}$). Among all peat samples,
the Alaskan peat fresh EF was the lowest EF (13.5±0.9 mg kg$^{-1}$), whereas Moscow peat fresh
emissions yielded the highest EF (69±4.4 mg kg$^{-1}$). The difference in total aromatic acid emissions
can be attributed to the variation in the lignin content of the fuels and burning conditions (Simoneit,

473   2002).







### 3.1.5 Resin Acids

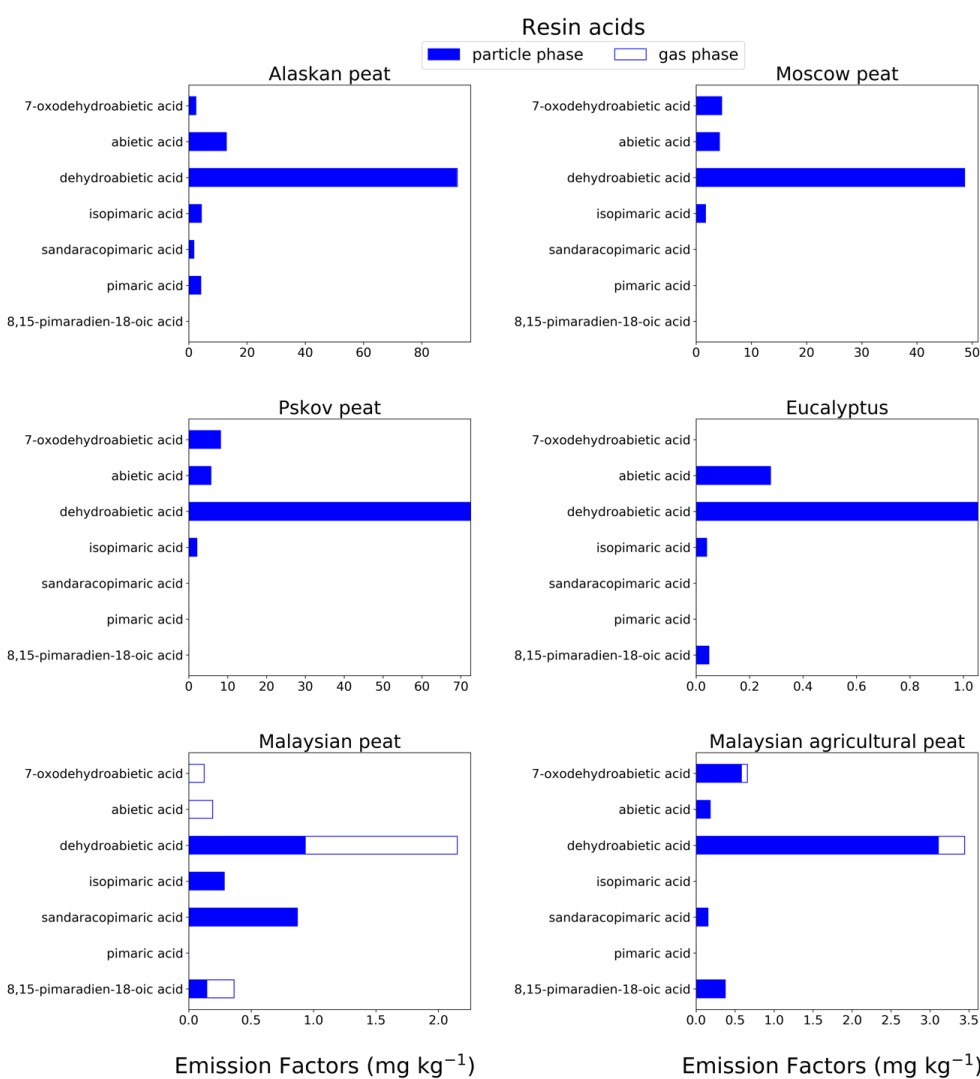


Figure 2e. EFs of resin acids in both particulate phase (solid bars, filter samples) and gas-phase
(open bars, XAD samples) from fresh biomass-burning emissions for six different biomass types.
As in prior cases, we did not burn fuels in replicates, and SD were calculated based on replicate
analysis of similar fuels (with same experimental conditions) from our previous campaigns





(Yatavelli et al., 2017a) and SD varies from 9.7–15% for resin acids
We quantitatively analyzed combustion emissions for isomers in six resin acids (Table S1). The
most abundant resin acid (78% of total resin acid emission) is dehydroabietic acid ($C_{20}$) that does
not have isomers. The preponderance of this acid over other resin acids in emissions from oak and
pine biomass burning was reported by Simoneit et al. (1993). We found that dehydroabietic acid
($C_{20}$) content in fresh emissions is 15–30 times higher in fuels from high-latitude peatlands than in
those of tropical origin. For example, the EF for dehydroabietic acid in fresh Alaskan peat
emissions is $92.2\pm14$ mg kg$^{-1}$ (Fig. 2e), whereas the same in fresh Malaysian peat emissions is
$3.44\pm0.5$ mg kg$^{-1}$ (Fig. 2e). Resin acids are supposed to be found mostly in particulate phase based
on their molecular weight and functional groups (Asher et al., 2002; Pankow and Asher,
2008)(Karlberg et al., 1988), confirmed by our results (80–100% in particulate phase) with the
exception of Malaysian peat emissions where 56.6% abundance of dehydroabietic acid was found
in gas phase. Although a distinct peak of dehydroabietic acid was observed at the desired retention
time for this sample during GC-MS analysis, we believe this result can be attributed to some
unknown interreference from our analysis procedure.

The EF$_{group}$ for seven resin acids are presented in Fig. 3e. High EF$_{group}$ was observed for Alaskan
($117\pm15$ mg kg$^{-1}$), Pskov ($89\pm12$ mg kg$^{-1}$), and Moscow ($59\pm7.7$ mg kg$^{-1}$) peats representing mid
latitude and arctic peats. Resin acids (e.g., pimaric acid) are biosynthesized mainly by conifers
(gymnosperms) in temperate regions. In previous work, Iinuma et al. (2007) gave a range of resin
acids EFs from 0 to 110 mg kg$^{-1}$, in agreement with our results. Very low resin acids EFs were
found for peat from tropical regions (e.g., 4 mg kg$^{-1}$ for Malaysian peat fresh samples). As
deciduous trees in tropical zones are not prolific resin and mucilage (gum) producers,
compositional data on smoke from such sources should not be expected to show moderate
concentrations of resin acids. This is supported by earlier work by Iinuma et al. (2007), where resin
acids were not even detectable for emissions from Indonesian peat combustion.





*3.1.5 Levoglucosan*
Levoglucosan can be found mostly in particulate phase (Simoneit, 2002). We report levoglucosan
EFs from our analysis of the quartz filter using the IC-PAD technique (no gas phase EFs reported).
The EFs of levoglucosan (Fig. 3f) were found to be $20.9\pm0.68$ mg kg$^{-1}$ and $485\pm11.8$ mg kg$^{-1}$ for
Eucalyptus and Malaysian peat, respectively, and their carbon content is approximately 1.8% and
2.5 % of the total organic carbon mass characterized by the thermo-optical technique. Fine et al.
(2002) reported 9% to 16% contribution of levoglucosan to total OC from residential wood
combustion, a relatively higher percentage than values obtained in our study. Anhydrosugars (e.g.,
levoglucosan and its isomers) are found in great abundance and have been widely used as a BB
tracer because of their atmospheric stability, as summarized by Bhattarai et al. (2019). We found
that levoglucosan constituted 36% and 51% of GC-MS characterized polar (listed in our method)
organic aerosol mass for eucalyptus and Malaysian peat, respectively, which also is consistent with
the previous BB literature assembled in the recent review article by Bhattarai et al. (2019).




**3.2. Emission factors of total (gas + particle) organic compounds of six chemical groups and**
**their changes upon OFR oxidations**

Here we describe changes in the $EF_{group}$ followed by OFR oxidation for all six chemical groups.
Levoglucosan and the most abundant resin acid, dehydroabietic acid also are reported in this
section.

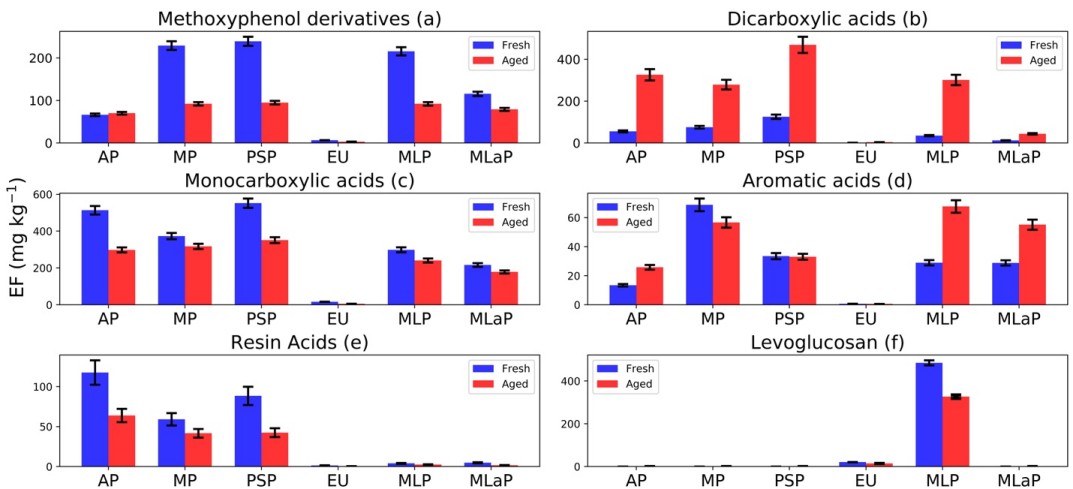

Figure 3. Fuel-based emission factors (EFs) of organic compounds assigned to six chemical groups
for six different fuels: Alaskan peat (AP), Moscow peat (MP), Pskov peat (PSP), Eucalyptus (EU),
Malaysian peat (MLP), Malaysian agricultural peat (MLaP). EFs are presented as a sum of gas-
and particle-phase species mass measured in fresh and OFR-aged BB emissions in units of mg kg⁻
¹ (mass of emissions per fuel mass combusted). We did not burn fuels in replicates, and standard
deviations (SD) of all chemical groups were calculated based on replicate analysis of similar fuels
(with same experimental conditions) from our previous campaign based on the data reported by
Yatavelli et al. (2017).

*3.2.1. Methoxyphenol derivatives after OFR oxidation (Fig. 3a)*
Methoxyphenols can undergo gas-phase oxidation reactions via either aromatic ring
fragmentation/opening to form short-chain ketones, acids, esters, and double bonds in conjugations





with all functional groups or further hydroxylation of aromatic rings to form multiple substituted
aromatic compounds (Yee et al., 2013). In either case, a decrease in methoxyphenols after
oxidation was expected. In our study, a decrease in methoxyphenol's $EF_{group}$ with OFR oxidations
were observed for all fuels (e.g., for Pskov peat from $239\pm11$ mg $kg^{-1}$ to $95\pm4$ mg $kg^{-1}$) except for
Alaskan peat, where an insignificant increase from $66\pm3$ mg $kg^{-1}$ to $70\pm3$ mg $kg^{-1}$ after the OFR
oxidation was observed.

*3.2.2. Dicarboxylic acid group after OFR oxidation (Fig. 3b)*
A significant increase (2.5–8.5 times) in the $EF_{group}$ of dicarboxylic acids was observed for OFR-
aged samples. For example, the $EF_{group}$ of dicarboxylic acids increased from $35\pm3$ mg $kg^{-1}$ to
$301\pm25$ mg $kg^{-1}$ for Malaysian peat and from $56\pm5$ mg $kg^{-1}$ to $326\pm27$ mg $kg^{-1}$ for Alaskan peat.
Oxidation of aerosols potentially produces more oxygenated functional groups (Jimenez et al.,
2009), demonstrated by an increase in O:C ratios (from 0.45 to 0.65) in recent laboratory oxidation
of BB emissions by Bertrand et al. (2018), where TAG-AMS was used to identify the fate of
organic compounds. In this work, however, the number of identifiable compounds with highly
functional groups is constrained by the elution technique used in the TAG method. Our results on
the fate of BB organic aerosols with 18 dicarboxylic acids can provide better mechanistic
understanding about the processes inside OFR.

*3.2.3. Monocarboxylic acid group after OFR oxidation (Fig. 3c)*
We observed a decrease in monocarboxylic acids $EF_{group}$ from OFR aging for all fuels. For
example, the $EF_{group}$ for monocarboxylic acids from Alaskan peat combustion decreased from
$514\pm23$ mg $kg^{-1}$ (fresh) to $298\pm14$ mg $kg^{-1}$ (aged). A relatively small decrease compared to Alaskan
peat was observed for Malaysian agricultural peat (from $216\pm10$ mg $kg^{-1}$ [fresh] to $179\pm8$ mg $kg^{-1}$
[aged]) and Malaysian peat (from $298\pm14$ mg $kg^{-1}$ [fresh] to $240\pm11$ mg $kg^{-1}$ [aged]) too. This is
probably because of the formation of low molecular weight (LMW) monocarboxylic acids (e.g.,
hexanoic acids; MW=116 g $mol^{-1}$) after OFR oxidation demonstrated in Fig. 4c and Table S2c and
will be discussed further in section 3.3. Monocarboxylic acids can be oxidized in the atmosphere
(Charbouillot et al., 2012), leading to the formation of dicarboxylic acids from $C_2$ to $C_6$ (Ervens et

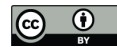



al., 2004). This is consistent with our results (Figs. 3b, 3c, Table S2b) Moreover, monocarboxylic
acids, during their atmospheric transformations, can produce a potential precursor for formation
of high molecular weight compounds, such as HUmic Like Substance (HULIS) (Carlton et al.,
2007; Tan et al., 2012).

*3.2.4. Aromatic acid group after OFR oxidation (Fig. 3d)*
Levels of aromatic acids from peat burning increased for Alaskan peat, Malaysian peat, and
Malaysian agricultural peat (e.g., from $29\pm2$ mg kg$^{-1}$ to $68\pm4$ mg kg$^{-1}$ for Malaysian peat) by OFR
aging (Fig. 3e, Table S2e). This increase could be from oxidation of phenols and methoxyphenols
in the OFR chamber (Akagi et al., 2011b; Legrand et al., 2016). For eucalyptus, Moscow and
Pskov peats it was an insignificantly small decrease (e.g., from $69\pm4$ mg kg$^{-1}$ to $57\pm3$ mg kg$^{-1}$ for
Moscow peat and from $34\pm2$ mg kg$^{-1}$ to $33\pm2$ mg kg$^{-1}$ for Pskov peat). This small decrease in the
$EF_{group}$ of monocarboxylic acids is statistically insignificant. The oxidation processes occurring in
the OFR are complex, especially in the case of multi-component BB emissions. The decrease
observed for aromatic acids after the OFR may be attributed, however, to multiple generations of
oxidation leading to the breaking of aromatic rings and formation of low molecular weight organic
compounds via fragmentation (Jimenez et al., 2009).


*3.2.5. Resin acids after OFR oxidation (Figs. 3d and 4e)*
Resin acids can be oxidized to corresponding oxo-acids (e.g., 7-oxodehydroabietic acid (Karlberg
et al., 1988)), and they are considered to be stronger contact allergens than the resin acids
themselves (Sadhra et al., 1998). Our data showed a small decrease in 7-oxodehydroabietic acid
levels after the OFR (e.g., from $8.2\pm0.8$ to $4.8\pm0.5$ mg kg$^{-1}$ for Pskov peat). We noted a significant
decrease in the $EF_{group}$ of resin acids from $117\pm15$ mg kg$^{-1}$ (fresh) and $63\pm8$ mg kg$^{-1}$ (aged) for
Alaskan peat after OFR oxidation, mostly because of some individual compounds like
dehydroabietic acid. Although resin acids are considered to be stable atmospheric tracers for
biomass burning (Simoneit et al., 1993a), we observed a decrease in dehydroabietic acid (most
abundant) EF after the OFR-oxidation of emissions from all fuels (Fig. S2b). For example, for





Alaskan peat (Fig. S2b), the decrease was from 92±14 mg kg$^{-1}$ to 57±8 mg kg$^{-1}$ after OFR-
oxidation. The fate of resin acids during OFR aging, however, was beyond the scope of this work
and may be the subject of future investigations.

*3.2.6. Levoglucosan after OFR oxidation (Fig. 3f)*
Levoglucosan is one of the most popular tracers of BB emissions, since it has been considered a
stable compound in the atmosphere (Oros et al., 2006; Simoneit, 2002; Simoneit et al., 1999).
Several laboratory studies, however, have demonstrated degradation of levoglucosan in the
presence of OH radicals (Hennigan et al., 2010). Here we observed a decrease of 30% in
levoglucosan levels following OFR oxidation. For example, Malaysian peat decreased from
485±12 mg kg$^{-1}$ to 327±8 mg kg$^{-1}$. For eucalyptus, the decrease was from 20±0.7 mg kg$^{-1}$ to 14±0.6
mg kg$^{-1}$. This decrease also can be attributed to the degradation process during OH oxidation
(Hoffmann et al., 2010). Levoglucosan oxidation should be studied more, so it can be adequately
used as a tracer of BB emissions.





**3.3. Aged-to-fresh ratios of total (gas + particle) emission factors of individual organic**
**compounds assigned to six chemical groups and their changes upon OFR oxidation**
We computed aged-to-fresh ratios of individual compounds for all fuels. If the aged-to-fresh ratio
of one compound is greater than one, this implies that the compound is formed during OFR
oxidations; if the ratio is less than one, then the compound must have decomposed inside the OFR.

*3.3.1 Methoxyphenol Derivatives*



Figure 4a. Aged-to-fresh ratios of total (gas + particle) EFs for methoxyphenols from biomass
burning emissions for six different biomass types presented in log scale. We did not burn fuels in
replicates, and standard deviations (SD) were calculated based on replicate analysis of similar fuels
(with same experimental conditions) from our previous campaigns. SD values derived from EFs
were scaled to ratio.

Overall, we found that abundances for methoxyphenol derivatives rapidly decreased upon OFR-
oxidation (Fig. 4a, Table S2a). Some compounds—vanillic acid, acetovanillone, and syringic
acids— demonstrated both increasing and decreasing trends. For example, for Pskov peat, the
aged-to-fresh ratio of guaiacol was 0.04±0.01 reflecting a significant decrease during OFR
oxidation. For Pskov peat, we also observed a ratio less than one for vanillin (0.44 ± 0.05),
indicating vanillin also decreased during OFR oxidation for the same fuel but not to the extent of
guaiacol. At the same time and for the same fuel, a slight increase (aged-to-fresh ratio >1) in
vanillic acid was observed (1.30±0.13) in the OFR-oxidized sample. This increase in vanillic acid
concentration can be attributed to the oxidation of vanillin, one of the abundant methoxyphenol in
the fresh emissions from Pskov peat (Fig. 4a, Table S2a). For combustion of other peats, vanillic
acid concentrations also decreased (e.g., aged-to-fresh ratios were 0.74±0.08 and 0.67±0.07 for
Alaskan peat and Malaysian agricultural peat, respectively). Acetovanillone increased by a factor
of three during OFR oxidation for Alaskan peat and around 15 % for Malaysian agricultural peat
(aged-to-fresh ratio 1.15±0.13), but the increase for Malaysian agricultural peat was not
statistically significant. For other fuels, acetovanillone decreased during the OFR oxidation. For
example, for Moscow Peat, the aged-to-fresh ratio for acetovanillone was 0.30±0.03. We still need
to investigate the reason why both acetovanillone and vanillic acid increased for some fuels and
decreased for others. The reduction of acetovanillone and vanillic acid was  because of a photo-
chemical decomposition process in the OFR with formation of lower molecular weight products,
such as succinic acid and maleic acid (Schnitzler and Abbatt, 2018).



*3.3.2 Dicarboxylic Acids*

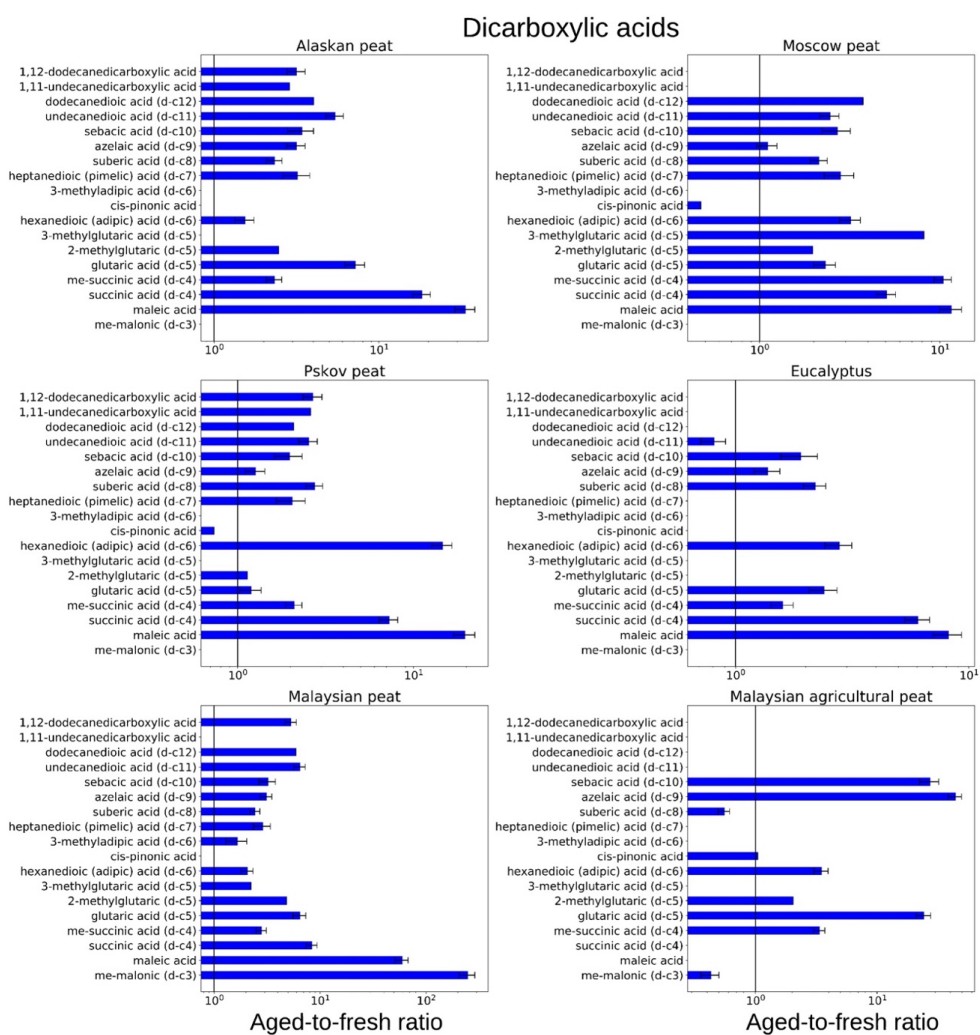


Figure 4b. Aged-to-fresh ratios of total (gas + particle) EFs for dicarboxylic acids from biomass
burning emissions for six different biomass types presented in log scale. We did not burn fuels in
replicates, and SD were calculated based on replicate analysis of similar fuels (with same
experimental conditions) from our previous campaigns. SD values derived from EFs were scaled
to ratio.





In the case of dicarboxylic acids, we observed 2–20 times increase in EFs, but the degree of
enhancement of low-MW (LMW) dicarboxylic acids EFs was higher than for high-MW (HMW)
dicarboxylic acids EFs. For example, a 20-fold increase in maleic acid, a LMW dicarboxylic acid
(MW= 116.07 g mol$^{-1}$), was observed during OFR oxidation of Pskov peat emissions (aged-to-
fresh ratio = 19.6±2.8), whereas 1,11-undecanedicarboxylic acid, an HMW dicarboxylic  acid
(MW = 244.33 g mol$^{-1}$), EF increased 2.6 times (aged-to-fresh ratio = 2.6±0.001) for the same fuel.
Similarly, the concentration of succinic acid, an LMW dicarboxylic acid (MW= 118.09 g mol$^{-1}$),
increased almost by five times after OFR oxidation (aged-to-fresh ratio = 5.07±0.62), whereas that
of undecanedioic acid, an HMW dicarboxylic acid (MW = 230.30 g mol$^{-1}$), increased 2.5 times
(aged-to-fresh ratio = 2.46±0.30) for Moscow peat. This trend was in accord with results from
ambient observations after BB  events (Cao et al., 2017; Kawamura and Bikkina, 2016).

















*3.3.3 Monocarboxylic Acids*

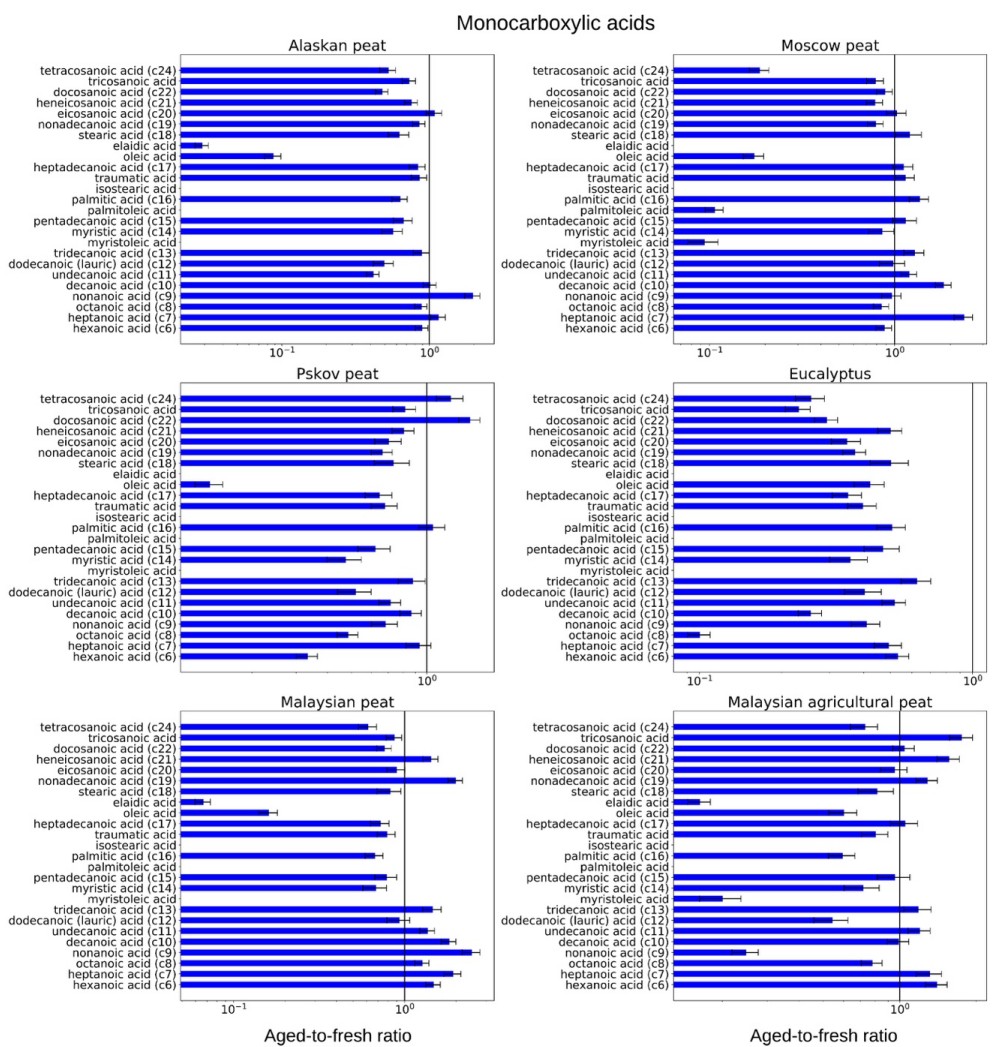


Figure 4c. Aged-to-fresh ratios of total (gas + particle) EFs for monocarboxylic acids from biomass
burning emissions for six different biomass types presented in log scale. We did not burn fuels in
replicates, and SD were calculated based on replicate analysis of similar fuels (with same
experimental conditions) from our previous campaigns. SD values derived from EFs were scaled
to ratio.



Our analysis of OFR-aged samples showed that concentrations of monocarboxylic acids with
different molecular weights changed during OFR oxidation, but the changes varied from one fuel
to another. For example, the EF of hexanoic acid ($C_6$) was reduced for Eucalyptus (aged-to-fresh
ratio = 0.5±0.05) and fuels from other high-latitude peatlands like Alaskan (aged-to-fresh ratio =
0.89±0.09), Moscow (aged-to-fresh ratio = 0.88±0.09) and Pskov peat (aged-to-fresh ratio =
0.33±0.03). The reduction of hexadecenoic acid was statistically insignificant for Alaskan and
Moscow peat. The peats from tropical regions showed exactly the opposite change. Hexanoic acid
increased for both Malaysian (aged-to-fresh ratio = 1.47±0.14) and Malaysian agricultural peat
(aged- to-fresh ratio = 1.40±0.14). We observed a similar trend for heptanoic acid ($C_7$) during OFR
oxidation. The tropical peats clearly demonstrated increases (for example, aged-to-fresh ratio =
1.92±0.22 for Malaysian peat) in heptanoic acid concentration. For Moscow peat, even though
hexanoic acid concentrations were insignificantly decreased, heptanoic acid concentrations
increased significantly (aged-to-fresh ratio = 2.37±0.27). This contrast between changes in
hexanoic and heptanoic acid can be explained by a decrease in CPI indices during OFR oxidations
(for example, from 2.78 to 1.7 for Malaysian peat). The reduction of CPI indices indicated that
during oxidation more monocarboxylic acids with odd carbon numbers were formed than
monocarboxylic acids with even carbon numbers. The abundance of hexadecenoic acid ($C_{16}$) was
reduced during OFR oxidation for all fuels (for example, aged-to-fresh ratio = 0.63±0.08 for
Alaskan peat) except for Moscow and Pskov peat aged-to-fresh ratio = 1.05±0.13 for peat), and
we believe this small increase is statistically insignificant. Similarly, tetracosanoic acid ($C_{20}$) was
reduced for all fuels (for example, aged-to-fresh ratio = 0.61±0.07 for Malaysian peat) except for
Pskov peat (aged-to-fresh ratio = 1.24±0.15). The small increase in tetracosanoic acid ($C_{20}$)
concentration was again statistically insignificant. Even though our results indicated the possibility
of fragmentation of HMW monocarboxylic acids and formation of LMW monocarboxylic acids,
this was because of the complexity of the OFR oxidation environment. We are not able to
hypothesize what is the main reactive mechanism.






**3.4. Most Contributing Compounds**

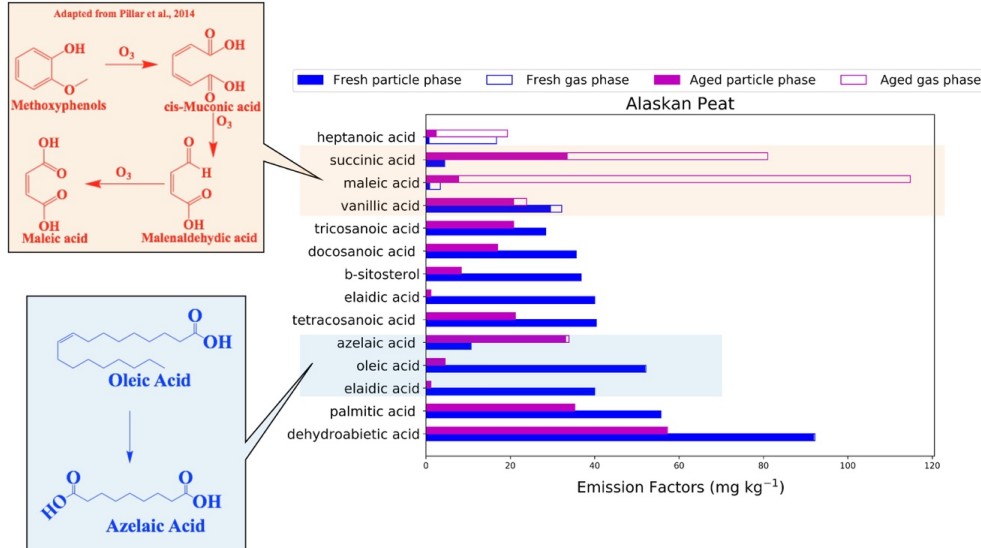



Figure 5. Emission factors (EF) of top contributing organic compounds assigned to Alaskan peat.
Top 10 contributing EFs were selected from both fresh and aged emissions. Since fresh and aged
samples do not have the same set of compounds after the selection, we included the top 10
compounds for both fresh and aged emissions. Hence, the number of top contributing compounds
varies from one fuel to another. Solid bars of each type represent a chemical group from particulate
emission of BB fuels and open bars of the same color represent gas phase BB emissions.

We have identified top contributing compounds for both fresh and aged BB emissions of each fuel
to understand how emissions vary from one fuel to another. The top 10 compounds for fresh and
aged emission were different, and we merged the top 10 compounds from fresh and aged emissions
resulting in different numbers of total top compounds for different fuels. Here we discuss Alaskan
peat emissions in their particulate phase (with solid bars) and gas phase (with open bars) as an
example, while the remaining results are given in the SM. It is clear that the top compounds vary
between fuels, likely because of the different chemical nature of these fuels.




Dehydroabietic acid, a resin acid, is the compound with the highest EF (92.2 mg kg$^{-1}$) for fresh
combustion emissions from Alaskan peat. Moncarboxylic acids including palmitic acid (EF = 55.7
mg kg$^{-1}$), tetracosanoic acid (EF = 40.35 mg kg$^{-1}$), and docosanoic acid (EF = 35.38 mg kg$^{-1}$) were
also found in high abundance in fresh emissions from the combustion of this fuel. The high
contributions of β-sitosterol (EF = 36.84 mg kg$^{-1}$) and alkenoic acids (e.g., oleic acid EF = 52.1
mg kg$^{-1}$) to emissions are unique to Alaskan peat. All the compounds described above are found
in particulate phase. After the OFR-oxidation, both dehydroabetic acid and β-sitosterol, considered
to be potential markers for biomass burning emissions (Simoneit et al., 1993b), decreased from
91.9 mg kg$^{-1}$ to 57.2 mg kg$^{-1}$  and 36.8 mg kg$^{-1}$ to 8.38 mg kg$^{-1}$ in particulate phase, respectively.
This reduction in EF because of OFR oxidation for both dehydroabetic acid and β-sitosterol must
be considered when using these compounds as biomass-burning markers. We observed the
formation of low molecular weight organic compounds, particularly in gas phase, from OFR
oxidation. For example, the EF of heptanoic acid increased from 2.42 mg kg$^{-1}$ to 16.9 mg kg$^{-1}$ and
that of maleic acid increased from 7.8 mg kg$^{-1}$ to 107 mg kg$^{-1}$ in the gas phase because of OFR
oxidation. Such a significant increase in  the EF of maleic acid can be explained by the aqueous
phase oxidation of methoxyphenols (El Zein et al., 2015) in the presence of ozone. We found that
the oxidation inside the chamber was happening under dry conditions and understand that the
reactions of organic compounds with OH radicals inside the OFR chamber will prevail over
reactions with ozone.  As we had our ozone scrubbers placed after sampling media (Fig. 1) to
prevent the pumps and online instruments from ozone-induced damage, we suspect that the maleic
acid was not formed inside the OFR chamber but rather by potential oxidation of organic
compounds on filters with relatively longer exposure of ozone (40–60 min for smoldering
combustion). Succinic acid EFs increased in both the gas phase (from 0.0 mg kg$^{-1}$ to 47.5 mg kg$^{-1}$
) and the particulate phase (from 4.43 mg kg$^{-1}$ to 33.5 mg kg$^{-1}$). Azelaic acid EFs showed mainly
an increase in the particulate phase (from 10.5 mg kg$^{-1}$ to 33.1 mg kg$^{-1}$), and we think that this was
because of the oxidation of oleic and eladic acid during OFR oxidation.






**3.5 Contribution of polar fraction to total organic carbon**


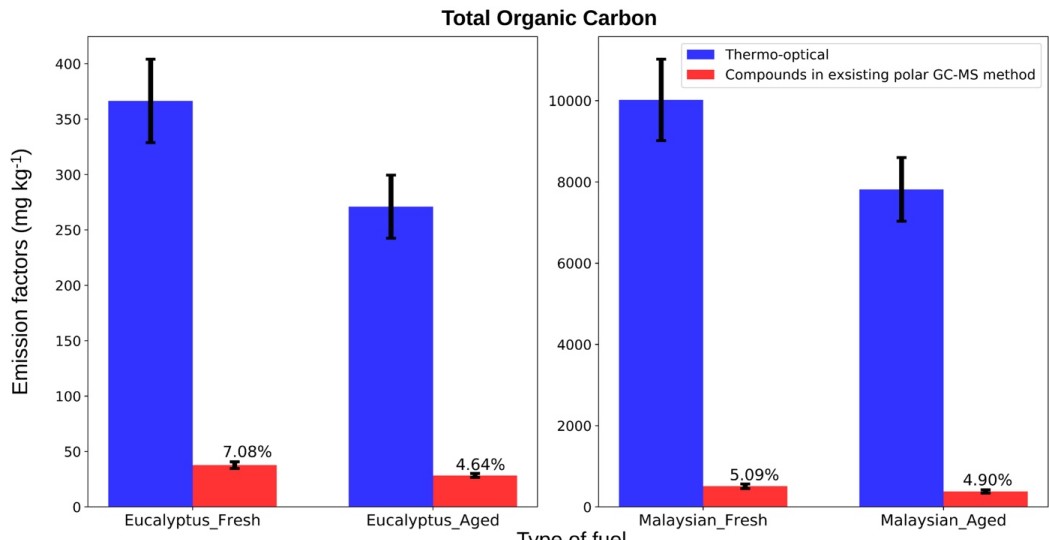


Figure 6. Contribution of GC-MS characterized polar compound carbon mass to total thermo-optical organic carbon mass. The y-axis shows the total carbon mass with dimensions of mass per mass. The error bars represent analytical uncertainties of the methods. For the thermo-optical method, uncertainties are the standard deviation of results from multiple punches on the same filter and for the GC-MS method, uncertainties were computed by taking the square root of sum of the squares of individual analytical uncertainties of all compounds included.

781

For Figure 6, we calculated the carbon content of total GC-MS characterized mass of identified polar organic compounds and compared results with the total OC mass characterized by the thermo-optical technique to estimate the contribution of polar compounds. The OC emissions were higher for smoldering combustion (10,209 ± 5 mg kg$^{-1}$ for Malaysian peat fresh emissions) than for flaming combustion (366.5 ± 7 mg kg$^{-1}$ for eucalyptus fresh emissions) samples, similar to the observation of flaming and smoldering combustion by Akagi et al. (2011). Total OC emissions are highly dependent on the type of fuel. For example, the fuel-based OC emission factor for rice crop residue burning is 1960 mg kg$^{-1}$ (Cao et al., 2008), whereas burning of corn and conifer forest



yields emission factors of ~1457 mg kg$^{-1}$ (Andreae and Rosenfeld, 2008) and ~7800 mg kg$^{-1}$
(Akagi et al., 2011a), respectively. Figure 6 shows that the 84 identified polar compounds in our
study constituted 4.5% to 7% of total OC mass for both fresh and aged emissions. From Indonesian
peat combustion emissions, Jayarathne et al. (2018) were able to identify polar compounds that
constituted 5.446% of total organic carbon mass. In recent work, based on both a field campaign
with prescribed burning and laboratory investigations, Jen et al. (2019) quantified a fraction (10–
65%) of only identified compounds (not a fraction of total mass) by the use of the 2D-GC-MS
technique. In our work, we identified only up to 7% (Fig. 6) of the total particle phase OC, and
further analysis of unidentified compounds is needed to improve understanding of atmospheric
chemistry of BB emissions.


## 3. Summary and conclusions

In this study, we chemically characterized the polar fraction of biomass-burning aerosols from
laboratory combustion of six different globally and regionally important fuels—five of them
representing smoldering and one of them representing flaming combustion. Our objective was to
understand how emissions of the polar compounds (e.g., methoxyphenols) varied from one fuel to
another during these combustion experiments and what are the relative distribution of these polar
compounds in gas and particulate phase. We also identified the fates of these polar compounds
following laboratory oxidation/aging (OFR aging). Resin acids were found mostly in emissions
from combustion of peats from high latitude regions but not in emissions of tropical peatlands
(e.g., EF$_{group}$ = 117±15 mg kg$^{-1}$ for Alaskan peat and EF$_{group}$ = 4.0±0.5 mg kg$^{-1}$ for Malaysian peat).
Similarly, monocarboxylic acids were found in higher abundance in emissions from high latitude
peatlands compared to tropical peatland emissions (e.g., EF$_{group}$ = 505±36 mg kg$^{-1}$ for Alaskan peat
and EF$_{group}$ = 212±15 mg kg$^{-1}$ for Malaysian agricultural peat). The presence of both guaiacol and
syringol moieties in all fuels indicated a part of the biomass, considered as representative of a
particular geographical region, is deciduous for all fuels. Low molecular weight compounds are
mostly found in gas phase (e.g., guaiacol found in gas phase 82–100%), whereas high molecular
weight (e.g., high molecular weight monocarboxylic acids [>C$_{16}$] more than 98% for all fuels) and
highly oxygenated compounds (e.g., syringic acid and acetovanillone 65–100% in particulate





phase) are found in particulate phase with high abundance. Monocarboxylic acids (1.2–3 times)
and methoxyphenols (1.5–2.5 times) decreased after OFR oxidation, whereas dicarboxylic acids
increased by 3–9 times followed by OFR oxidation. Relatively low molecular weight hexanoic
acid increased for both Malaysian (aged-to-fresh ratio = 1.47±0.14) and Malaysian agricultural
peat (aged- to-fresh ratio = 1.40±0.14), whereas high molecular weight tetracosanoic acid ($C_{20}$)
was reduced for all fuel increases (e.g., aged-to-fresh ratio = 0.61±0.07 for Malaysian peat). This
indicated fragmentation occurring inside the OFR chamber. With relative distribution of the top
10–15 compounds from Alaskan peat, we were able to identify transformation of unsaturated fatty
acids (e.g., oleic acid) to dicarboxylic acids (e.g., azelaic acid). We identified only up to 7% of the
total particle phase OC and further analysis of unidentified compounds with GC-MS full scan is
needed for better understanding of atmospheric chemistry of BB emissions.


**Data availability.** Data can be provided upon request: <andrey.khlystov@dri.edu>.

**Author contributions.** DS, VS, and AK designed experiments. DS and CB performed sample and
data collection. DS performed extractions, derivatizations, GC-MS analysis, summarized data, and
wrote the paper. AW provided biomass fuels. VS, AK, and HM provided input on interpretation
of results. VS, HM, and AK revised the manuscript.

**Competing interests.** The authors declare that they have no conflict of interest.

**Acknowledgements.** This research was supported by the National Science Foundation (NSF)
under grant numbers AGS-1544425 and AGS-1408241, NASA ROSES under grant number
NNX15AI48G, and internal funding from DRI. The authors would like to thank Anna Tsibar
(Moscow State Lomonosov University, Moscow, Russia) for providing peat fuels from Russia.
We acknowledge Benjamin Nault (CIRES, UC Boulder) and Andrew Lambe (Aerodyne) for their
insightful discussion leading to identification of a potential artifact in our experimental set up
associated with high maleic acid formation. The authors also thank Rodger Kreidberg for revising
the manuscript.





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
