# Peer review of "Polar semi-volatile organic compounds in biomass burning emissions and their chemical transformations during aging in an oxidation flow reactor"

_Atmospheric Chemistry and Physics, 2019_

## Referee Comment (RC1) · Anonymous Referee #2 · 18 Feb 2020

**Review of acp-2019-1179**

Review

February 18, 2020

**1 General Comments**

This paper uses a flow tube reactor to simulate the atmospheric oxidation potential of organic molecular markers emitted from burning peat and eucalyptus. The peat fires smoldered while the eucalyptus ones flamed. The semi-volatile organic markers were measured off-line using GC-MS or IC-PAD. The study was able to identify the reaction potential and the formation of semi-volatile organic markers that are common to chemical mass balance models and source apportionment studies.

The study topic is good for *ACP*. The results are timely because oxidation flow reactors (OFRs) are coming into vogue, and we need to better understand the results that OFRs produce and the potential air quality implications. With that, the study data have value and should be published. However, this particular study also has some serious limitations that need to be better addressed.

For one, the paper organization makes it seem like there are two separate studies, an emissions study and an OFR study. The emissions part of the paper is substantially less useful because all of the semi-volatile markers were observed in past studies and reported about repeatedly. Additionally, the discussion on emissions lacks information about how these peat burning results are different than other forms of vegetation burning and what that may or may not mean for emissions reactivity. More critical thinking is warranted to help emphasize how the peat burning emissions are going to impact the SOA compared with other biomass fuels. Otherwise, it does not belong at the forefront of the paper currently. The best approach may be to emphasize the emissions work less and get to the OFR results sooner.

The lack of replicate tests is another serious study limitation. It appears that standard deviations were taken from another set of tests and applied. This makes it difficult to judge the quality of the concentration data given here and calls into question how exactly the statistics were performed. It also complicates the interpretation of evidence showing what reacted or formed. This is a major limitation that needs to be addressed by providing the readers with sample population ($N$) and a detailed description of how the standard deviation was calculated and used to determine statistical significance and so on.

Finally, no new emissions compounds or SOA markers were measured that weren't part of the original compound suite. At the very least a set of chromatograms should be provided showing the raw emissions and the OFR effluent. That way we can see the major changes in these emissions.

More specific comments that may help remedy these major deficiencies are given below.

**2 Specific comments**

1. line 13: conserve use of the term 'significant' for describing statistics results.

2. line 48: Biomass burning particles exhibit a variety of toxicological properties other than mutagencity. It is worth being clear about that.

3. lines 54-59: **[MAJOR]** Resolving the organic chemical composition of biomass burning particles is indeed challenging and improvements in speciation are needed. However, this particular study doesn't improve speciation technology in any way. It examines a common set of polar organic compounds and simulates their potential to photooxidize in the atmosphere. The suggestion is to focus on the reactivity of these molecules not on the improved speciation seeing that there are no methodological improvements in molecular level speciation being presented *per se*. After this study, we are still near 80% w/w for the unknown chemical fraction.

4. lines 61-72: This study uses peat as the main BB fuel. Be more specific about the importance of peat burning and why there needs to be a global emphasis on peat burning emissions.

5. lines 80:81: I think the paper needs to be written around this sentence. This is what will make this study novel, important, and worth reading to many. Try and get to this point sooner. It would be better to couch this paper as one that attempts to examine the stability of commonly used polar atmospheric organic markers in an oxidizing environment.

6. lines 81-93: Describe the advantages and disadvantages of each of these studies, how these pertain and give rise to the current study. Describe more than what these studies did, i.e, mention why these studies are important and are worthy of further discussion. Otherwise, there is not much value here.

7. lines 95-96: All peats except Eucalyptus? Why not just focus on peats?

8. lines 97-98: Describe why the focus here was isolated to polar compounds. It may have been more interesting to focus on the most reactive compounds in the aerosol mixture instead of presenting the study as an exercise in chemical accounting.

9. lines 135-140: At what depth were the tropical and Russian peats extracted?

10. lines 176-179: Briefly mention the mass of fuel used per test and how the fuels were configured for burning.

11. lines 203-207: Figure 1 does not indicate which on-line instruments were used. These instruments should be described. Please explain how they were used. If they aren't being used as part of this study please explain why. There is no modified combustion efficiency evidence showing the levels of smoldering or flaming combustion for each fuel type. Perhaps that can be included here.

12. lines 229-231 and lines 251-258: It is mentioned that a $^{13}C$ radio-labeled levoglucosan sample was added as internal standard. Wouldn't the $^{13}C$ isotope elute at the same time as the unlabeled compound? How was that accounted for in this study? The radio isotopes are typically applied for GC-MS. How did things work with the IC-PAD method knowing that the PAD is a non-specific detector and that the radio isotope may interfere?

13. lines 236-237: What is the value of reporting these separately considering that the equilibrium partitioning changes and was specific to the sampling conditions used for this particular study? Please discuss briefly. Also, was any thermodynamics-based check performed on how well the equilibrium partitioning was measured here? Such a relatively straightforward check on select compounds can verify that the sampling worked as planned.

14. lines 275-280: **[MAJOR]** Please clarify how the SDs were calculated. It's unclear where the standard deviations (SDs) came from and the reader should know exactly what and how many experiments were used to report SD values. The suggestion that the SDs came from another study and therefore can be applied here is questionable analytical-chemical practice. SDs should be experiment-specific, not just assumed and carried-over from study to study. At the very least, multiple injections of the same extract should be performed. What is meant by 'similar' fuels? This comment applies to all of the experimental data being reported.

15. lines 282-328: This information about methoxy-phenol emissions from biomass burning is already available *ad naseum* in the literature and can be further consolidated. Instead of focusing on what we already know, examine how the peat emissions are different (if they indeed are) and how these differences may be important to the atmospheric processing and oxidation. Knowing more about the extent of smoldering and flaming combustion can help properly develop this discussion.

16. lines 381-508: **[MAJOR]** Again, virtually all of this chemical information is already available. Much of this can be condensed into a brief paragraph or two unless there is something salient and different for the fuels being examined here that merits further attention. **Suggestions:** Develop one section that discusses 'Polar organic chemistry' of these emissions. Move the figures of individual compound emissions into supporting information. Develop a new emission figure that combines everything by compound class and shows the reader some sort of chemical mass balance so they can understand the aerosol fraction of interest (more like Figure 3, but include a mass balance). Only if needed, focus on some of the unique chemical or physical state attributes of individual compounds emitted during burning of these particular fuels. In other words, if the chemical structure or physical state of a compound is important to understanding its oxidation then discuss these attributes and the results. This requires a reorganization of the paper as some of these results are presented elsewhere.

17. line 509: It would be interesting to learn if any of the levoglucosan was found in the gas phase. Levoglucosan elutes from a GC column with and without derivatization. Was there any indication of levoglucosan in the gas-phase?

18. lines 524-530: The important part of the study begins here. Figure 3 is a good figure that captures the essence of what was tested, measured, and and accomplished.

19. Figure 3: Again, are these error bars taken from other studies? If so, please remove them from the figure. One idea is to combine all of the peat samples measured here and calculate a standard deviation and use that to report global error for experimental peat burning. That can be added as a panel to one of the figures or presented separately.

20. lines 605-614: The importance of associating meaningful error with concentration data cannot be understated. For example, the error associated with levoglucosan measurement using GC-MS is approximately 20%. Although, the combustion test error can bring this value closer to 30%-50%. If we knew the exact error associated with repeating these peat experiments, we'd have a better understanding of this decrease being real or not. This may also explain why the same compound appears to be either formed or degraded for some fuels (see lines 631-632 and lines 691-699 (*e.g.*, hexanoic acid).

21. line 696: Please describe the type of statistical tests being applied here and how they are being applied. Please report the sample population (*N*). Additionally, provide more information about how the error was calculated and applied to conduct the statistics tests.

22. lines 724-733: Please describe the criteria for a "top" contributing compound. Was it concentration?

23. lines 736-737: Isn't it also possible that the combustion was different?

---

## Referee Comment (RC2) · Anonymous Referee #1 · 22 Mar 2020

This study quantified a number of compounds produced by burning six different fuels, most of them peat, and examined the effect of aging of the biomass burning smoke by reactions with OH (and to some extent ozone) in an oxidation flow reactor (OFR). The authors report emission factors (EFs) for all the compounds as well as the change in the effective EFs after OFR.

Major comments 1. The reported dataset is comprehensive but it is not very clear to the reader what one can do with these data. While the decrease in the EFs can likely be modeled with known OH rate constants, the increasing EFs are hard to interpret

without knowing whether additional products came from gas-ozone chemistry or from heterogeneous oxidation of particles by OH. There are some hints of this discussion in the paper, for example, on line 554. It would be good to expand this discussion. 2. I always find it surprising how small the mass fraction of quantified compounds is despite clearly sophisticated analytical chemistry methods. Would it be possible to discuss what the rest of the compounds could be? Are there giant peaks in the GC data that are not reported because of the lack of standards?

Minor comments Lines 126-133, 144-154: These sentences belong to either introduction or discussion. In the experimental section, it is better to focus on the specifics of the fuels used (such as water content, carbon content, etc.), not on the general importance of peatland ecosystems and eucalyptus forests. Line 165: it would probably help to know which molecule comes from which company Line 215: I do not think it is a sufficient argument. Reactions with OH are limited to a short time the mixture spends in OFR. In contrast, residual ozone comes out from OFR and has a chance to react with compounds on filter for the duration of sampling. Therefore, secondary oxidation of BB compounds on filter by ozone is quite possible and can only be ruled out if the residual ozone is low. The authors discuss these effects later in the paper on line 760. Line 287: MW should be 124.14 g/mol Line 360: missing right parenthesis Line 492: replace )( with a semicolon Lines 568, 660: you already used LMW on line 448, it would be better to define it there, or not use this abbreviation at all, as it is used only a few times Line 664: error in 2.4+/-0.001 appears to be unrealistically small considering other errors reported in this paper Figure 4c: two panels have only one number on the X-axis, and ticks are too small making it hard to estimate the values from the figure (the tick comments applies to all similar figure). Figure 5: the text (on line 736) suggests that similar figures exists for other fuel types, however, they are not presented in the supporting information section
* * *

---

## Author Comment (AC1) · 7 May 2020

Reviewer 1: This paper uses a flow tube reactor to simulate the atmospheric oxidation potential of organic molecular markers emitted from burning peat and eucalyptus. The peat fires smoldered while the eucalyptus ones flamed. The semi-volatile organic markers were measured off-line using GC-MS or IC-PAD. The study was able to identify the reaction potential and the formation of semi-volatile organic markers that are common to chemical mass balance models and source apportionment studies. The study topic is good for ACP. The results are timely because oxidation flow reactors (OFRs)

[Figure]

are coming into vogue, and we need to better understand the results that OFRs produce and the potential air quality implications. With that, the study data have value and should be published. However, this particular study also has some serious limitations that need to be better addressed.

AC: We thank the reviewer for the detailed and constructive review of our manuscript. Below, we first respond to general comments and then we address specific comments. All corrections were highlighted in the marked manuscript (Sengupta_manuscript_ACPD2019_review_marked_Final).

General comments:

Reviewer 1: For one, the paper organization makes it seem like there are two separate studies, an emissions study and an OFR study.

AC: We recognize the reviewer's concern and agree that the study does consist of two main parts (discussion of fresh and aged BB emissions). Due to the complexity of the data (EFs: fresh vs. aged, gas vs. particulate phase emissions from six different fuels) we prefer to preserve the current structure of the manuscript. Initially, we intended to present our results for fresh and aged emissions together. However, in order to increase coherency in narration we described the fresh emissions and corresponding changes after OFR oxidation separately.

Reviewer 1: The emissions part of the paper is substantially less useful because all of the semivolatile markers were observed in past studies and reported about repeatedly. Additionally, the discussion on emissions lacks information about how these peat burning results are different than other forms of vegetation burning and what that may or may not mean for emissions reactivity. More critical thinking is warranted to help emphasize how the peat burning emissions are going to impact the SOA compared with other biomass fuels. Otherwise, it does not belong at the forefront of the paper currently. The best approach may be to emphasize the emissions work less and get to the OFR results sooner.

AC: We agree with the reviewer that some of the semivolatile markers described in this study have been already presented in other studies (Mazzoleni et al., 2007; Yatavelli et al., 2017). On the other hand, we are convinced that our research provides new and very important data on fresh emissions from globally important biomass fuels, which have not been characterized extensively before. We added the importance of the emissions from analyzed fuels in the "Introduction" section (Pages 3-4, lines 71-79). The goal of this study was not to identify new BB organic compounds in fresh and aged emissions. Our research was mainly focused on analysis of organic compounds (with well-developed methods) and compare them between fresh and aged BB emissions from different fuels. In addition to polar organic compounds we also analyzed PAHs, alkanes, cycloalkanes, etc. Moreover, our next goal is to identify and quantify new BB compounds. These data will be presented in our forthcoming papers. We agree with the reviewer that it would be better if the OFR results appear sooner in the manuscript, however, as we mentioned in our previous comment, because of the inherent complexity of the data we are inclined to keep the current structure of our manuscript.

Reviewer 1: The lack of replicate tests is another serious study limitation. It appears that standard deviations were taken from another set of tests and applied. This makes it difficult to judge the quality of the concentration data given here and calls into question how exactly the statistics were performed. It also complicates the interpretation of evidence showing what reacted or formed. This is a major limitation that needs to be addressed by providing the readers with sample population (N) and a detailed description of how the standard deviation was calculated and used to determine statistical significance and so on.

AC: We recognize the lack of replicate analysis is a limitation of the current study. Unfortunately, due to resource (limited quantity of fuels available) and budget limitations of the project, we were able to perform only single-burning experiments with each fuel. However, in our previous BB experiments which were run with comparable conditions (Samburova et al., 2016; Yatavelli et al., 2017), we were able to perform triplicate burns

for each fuel (including peats) and we argue that the uncertainties from our previous campaigns can be used for the current burning experiments. A detailed description was added to the section 2.5 The uncertainty of the emission factor values reported in the current manuscript includes: 1) burn to burn variability, 2) wall losses in the OFR, and 3) analytical uncertainty of the GC-MS method. In our previous campaigns, we clearly observed that the burn-to-burn variability is much higher than the wall losses and the analytical uncertainty of our method. There is no reason to believe that burn-to-burn variability, as well as other uncertainties in the current study are different from those observed in our previous studies. Therefore, we think that the best approach is to use the levels of uncertainty that we calculated for our previous combustion campaigns (calculated based on three replicate burns where a comparable experimental set up was used) and use these values for the present data set. For levoglucosan analysis with IC-PAD technique, we were able to calculate only analytical uncertainty.

Reviewer 1: Finally, no new emissions compounds or SOA markers were measured that weren't part of the original compound suite. At the very least a set of chromatograms should be provided showing the raw emissions and the OFR effluent. That way we can see the major changes in these emissions.

AC: We thank the reviewer for this proposition. We could not agree more that finding new SOA markers is really important in this area of atmospheric research. As mentioned in the current manuscript, we actually performed GC-MS full scan runs on our samples and we are currently working on the data for a future publication discussing compounds that fall outside of the current method.

Specific comments:

1. line 13: conserve use of the term 'significant' for describing statistics results.

AC: We changed "significant" on " prominent" (Page 1, Line 12). The word "significant" related to statistical results was also reviewed and changed through the text

2. line 48: Biomass burning particles exhibit a variety of toxicological properties other than mutagencity. It is worth being clear about that.

AC: We agree, BB particle and gas exhibits wide toxicological properties and mutagenicity is one of them. Hence, we have rephrased the sentence and added a few more refences on the toxicity of BB aerosols (Page 3, lines 47 – 49)

3. lines 54-59: [MAJOR] Resolving the organic chemical composition of biomass burning particles is indeed challenging and improvements in speciation are needed. However, this particular study doesn't improve speciation technology in any way. It examines a common set of polar organic compounds and simulates their potential to photooxidize in the atmosphere. The suggestion is to focus on the reactivity of these molecules not on the improved speciation seeing that there are no methodological improvements in molecular level speciation being presented per se. After this study, we are still near 80% w/w for the unknown chemical fraction.

AC: We appreciate reviewer's point and agree that this study is not novel in identification of new BB species. The goal of this research was to compare fresh and OFR aged BB emissions of known compounds, which have been already identified in previous studies. We deleted this part of the text from the main manuscript to avoid a wrong impression that we also characterized unknown BB compounds (Page 3, lines 55-60). Also, we added a sentence (Pages 5-6, lines 132-135 of the marked manuscript) that clarifies the goal of this study.

4. lines 61-72: This study uses peat as the main BB fuel. Be more specific about the importance of peat burning and why there needs to be a global emphasis on peat burning emissions

AC: This is a good recommendation. Analysis of both fresh and aged peat burning emissions is indeed one of the strengths of this work. We added a paragraph that explains the importance of the peat emissions (Pages 3-4, lines 71-79).

5. lines 80:81: I think the paper needs to be written around this sentence. This is what will make this study novel, important, and worth reading to many. Try and get to this point sooner. It would be better to couch this paper as one that attempts to examine the stability of commonly used polar atmospheric organic markers in an oxidizing environment.

AC: We do agree that in our paper we attempted to investigate the stability of commonly identified polar organic compounds in laboratory oxidizing environment. However, we also wanted to analyze and discuss the difference in EFs of these compounds between different fuels as total (gas + particulate phases) and as well as individual gas and particulate phases.

6. lines 81-93: Describe the advantages and disadvantages of each of these studies, how these pertain and give rise to the current study. Describe more than what these studies did, i.e, mention why these studies are important and are worthy of further discussion. Otherwise, there is not much value here.

AC: This is a great suggestion. We added both advantages and disadvantages of the studies to the manuscript (Pages 4-5, lines 101-113).

7. lines 95-96: All peats except Eucalyptus? Why not just focus on peats?

AC: We understand the reviewer's recommendation to focus on peat fuels only. However, Eucalyptus fuel is ubiquitous and has a large contribution to recent Australian wildfire emissions. Therefore, presenting our results for eucalyptus emissions is timely and we prefer to keep them in our manuscript with respect to global emission perspectives. The importance of eucalyptus is now highlighted in section 2.1 (Page 7, lines 167 -171)

8. lines 97-98: Describe why the focus here was isolated to polar compounds. It may have been more interesting to focus on the most reactive compounds in the aerosol mixture instead of presenting the study as an exercise in chemical accounting.

AC: We understand the reviewer point. 84 polar compounds were selected and analyzed based on our established "polar compound" method, since there are no data how these compounds behave during OFR-aging in the complex mixture of real BB emissions.

9. lines 135-140: At what depth were the tropical and Russian peats extracted?

AC: The peat soils were extracted from the top 15cm of the soil profile, in accordance with convention and also to represent the depths at which soil combustion is most likely to occur. More details were added to the Experimental section (Page 7, lines 163-165)

10. lines 176-179: Briefly mention the mass of fuel used per test and how the fuels were configured for burning.

AC: Masses of all fuels were added to the Experimental section and the description of fuels' configuration during burning experiments was also added (Page 7, lines 179-180)

11. lines 203-207: Figure 1 does not indicate which on-line instruments were used. These instruments should be described. Please explain how they were used. If they aren't being used as part of this study please explain why. There is no modified combustion efficiency evidence showing the levels of smoldering or flaming combustion for each fuel type. Perhaps that can be included here.

AC: The following on-line instruments were used during our combustion experiments: Scanning Mobility Particle Sizer (SMPS), Photo Acoustic Soot Spectrometer (PASS-3), CO and CO2 analyzer. More detailed description of the instrumental set-up can be found in our previous publication (Bhattarai et al., 2018). PASS and SMPS results will be used in our following manuscript, which will focus on optical properties of BB emissions. During this campaign, the CO2 analyzer malfunctioned and thus we were unable to use/present the continuous CO2 data and hence the modified combustion efficiency.

12. lines 229-231 and lines 251-258: It is mentioned that a 13C radio-labeled levoglucosan sample was added as internal standard. Wouldn't the 13C isotope elute at the same time as the unlabeled compound? How was that accounted for in this study. The radio isotopes are typically applied for GC-MS. How did things work with the IC-PAD method knowing that the PAD is a non-specific detector and that the radio isotope may interfere?

AC: Thank you for catching this error. Levoglucosan-d7 was used as an internal standard for the quantitative analysis of levoglucosan with GC/MS. For the IC-PAD analysis of levoglucosan (see comment #17) the filters were not spiked with any internal standards. This error was corrected (Page 8, lines 194-195).

13. lines 236-237: What is the value of reporting these separately considering that the equilibrium partitioning changes and was specific to the sampling conditions used for this particular study? Please discuss briefly. Also, was any thermodynamics-based check performed on how well the equilibrium partitioning was measured here? Such a relatively straightforward check on select compounds can verify that the sampling worked as planned.

AC: This is definitely an interesting scientific question. We agree that the equilibrium partitioning can change under different sampling conditions. However, a compound's phase also affects its reaction rates. Thus, we believe it is important to report gas and particle concentrations as they were observed during the experiments. We did check whether the observed partitioning is thermodynamically reasonable – it is for most compounds, at least qualitatively. A quantitative assement, however, is complicated for the heavy and light compounds due to the very low concentrations in one of the phases. We are working on resolving these issues, but the complexity of this process is well beyond the scope of this manuscript.

14. lines 275-280: [MAJOR] Please clarify how the SDs were calculated. It's unclear where the standard deviations (SDs) came from and the reader should know exactly what and how many experiments were used to report SD values. The suggestion that

the SDs came from another study and therefore can be applied here is questionable analytical-chemical practice. SDs should be experiment-specific, not just assumed and carried-over from study to study. At the very least, multiple injections of the same extract should be performed. What is meant by 'similar' fuels? This comment applies to all of the experimental data being reported.

AC: In the section 2.5 we added a description how the SDs were calculated for the present data set (Page 11, lines 280 – 293). In our analytical method we have replicates for few samples with multiple injections that we have used for calculating analytical uncertainties. However, burn-to-burn variability during the combustion experiments governs the overall uncertainties and hence we have included that variability calculated from the data collected in our previous combustion campaigns. It is worth mentioning that analytical uncertainties are also discussed along with burn-to-burn variability. The uncertainties for the peat combustion experiments were computed based on the results for Alaskan peat emissions (triplicate burns), while the uncertainties for eucalyptus were calculated using our previous Cheatgrass results (triplicate burns). The expression "Similar fuels" refers to similar combustion conditions (smoldering type peat combustion vs. flaming type eucalyptus combustion) (Page 13, line 320)

15. lines 282-328: This information about methoxy-phenol emissions from biomass burning is already available 'ad naseum' in the literature and can be further consolidated. Instead of focusing on what we already know, examine how the peat emissions are different (if they indeed are) and how these differences may be important to the atmospheric processing and oxidation. Knowing more about the extent of smoldering and flaming combustion can help properly develop this discussion.

AC: We have this discussion (differences in EFs between fuels) in our manuscript and we also compared our EFs with EFs from other studies. (Section 3.1.1).

16. lines 381-508: [MAJOR] Again, virtually all of this chemical information is already available. Much of this can be condensed into a brief paragraph or two unless there is

something salient and different for the fuels being examined here that merits further attention. Suggestions: Develop one section that discusses 'Polar organic chemistry' of these emissions. Move the figures of individual compound emissions into supporting information. Develop a new emission figure that combines everything by compound class and shows the reader some sort of chemical mass balance so they can understand the aerosol fraction of interest (more like Figure 3, but include a mass balance). Only if needed, focus on some of the unique chemical or physical state attributes of individual compounds emitted during burning of these particular fuels. In other words, if the chemical structure or physical state of a compound is important to understanding its oxidation then discuss these attributes and the results. This requires a reorganization of the paper as some of these results are presented elsewhere.

AC: We really appreciate reviewer's recommendation of creating a new section and figures that exhibit a mass balance. We have addressed that issue in the general comment section too. Here, we give an example that explains why attaining closure in mass balance with the our results is a difficult task and beyond the scope of this manuscript. For example, aromatic acids can be produced during oxidation of methoxyphenols (reported in this study) and also from oxidation of PAHs and substituted PAHs (Wang et al., 2007) (not reported). Similarly, monocarboxylic acids can form by fragmentation (during oxidation) of compounds, form the same homologous series, and at the same time they also can be derived from oxidations of alkanes and alkenes. Because of the chemical complexity, it is very challenging to derive adequate mass balance results based only on analyzed polar compounds without considering other species.

17. line 509: It would be interesting to learn if any of the levoglucosan was found in the gas phase. Levoglucosan elutes from a GC column with and without derivatization. Was there any indication of levoglucosanin the gas-phase?

AC: For the quantitative analysis of levoglucosan we selected IC-PAD analysis over the GC-MS technique that includes BSTFA-derivatisation of this polar compound. Due to the high levels of levoglucosan in the extracts, the MS detector was oversaturated;

that made it impossible to perfom adequate quantitative analysis of this compound. Therefore, further investigation of levoglucosan in the gas phase emissions was not feasible.

18. lines 524-530: The important part of the study begins here. Figure 3 is a good figure that captures the essence of what was tested, measured, and and accomplished.

AC: We thank the reviewer for this recognition

19. Figure 3: Again, are these error bars taken from other studies? If so, please remove them from the figure. One idea is to combine all of the peat samples measured here and calculate a standard deviation and use that to report global error for experimental peat burning. That can be added as a panel to one of the figures or presented separately. AC: The uncertainties calculations are added to the text (Page 11, lines 280-293)

20. lines 605-614: The importance of associating meaningful error with concentration data cannot be understated. For example, the error associated with levoglucosan measurement using GC-MS is approximately20%. Although, the combustion test error can bring this value closer to 30%-50%. If we knew the exact error associated with repeating these peat experiments, we'd have a better understanding of this decrease being real or not. This may also explain why the same compound appears to be eitherformed or degraded for some fuels (see lines 631-632 and lines 691-699 (e.g., hexanoic acid).

AC: This is an interesting point. However, we believe that this is not only due to the error assignment. Hexanoic acid can be formed by oxidation of alkanes and decomposition of high molecular weight mono carboxylic acids (McNeill et al., 2008) This is why it is difficult to predict the fate of this compound by just looking at a matrix of polar compounds. Moreover, the occurance of multiple generation of oxidations inside OFR (Li et al., 2015) leads to further complexity and as a result the same compounds can be either formed or degraded during oxidations of emissions from different fuels.

21. line 696: Please describe the type of statistical tests being applied here and how they are being applied. Please report the sample population (N). Additionally, provide more information about how the error was calculated and applied to conduct the statistics tests.

AC: We have described earlier how we calculated SD at the end of section 2.5 (Page 11, lines 280-293). In order to evaluate changes associated with oxidation experiments, we need to perform "Pairwise-T-test" with fresh and aged samples. However, in our case we have computed the uncertainties from our previous campaign with only fresh samples, propagated that to both fresh and aged samples, and we refrained from performing pairwise T-test for this analysis. We agree with the reviewer's perspective that without replicate burns, we need to be conservative while describing some change as "statistically significant/insignificant". We have deleted such strong words to make the message less categorical. Changes in the main manuscript related to this comment are highlighted (Lines 621, 681, 733, 747, 750).

22. lines 724-733: Please describe the criteria for a "top" contributing compound. Was it concentration?

AC: The reviewer is correct, the "top" contribution compounds refer to compounds with highest levels of concentration (EFs.). We rephrased the figure caption (Page 35, lines 762-763)

23. lines 736-737: Isn't it also possible that the combustion was different?

AC: We agree with the reviewer and added the "nature of combustion" to the text (Page 36, line 776).

References:

Bhattarai, C., Samburova, V., Sengupta, D., Iaukea-Lum, M., Watts, A.C., Moosmüller, H. and Khlystov, A.Y., 2018. Physical and chemical characterization of aerosol in fresh and aged emissions from open combustion of biomass fuels. Aerosol Science and

[Figure]

Technology, 52(11), pp.1266-1282.

Li, R., Palm, B. B., Ortega, A. M., Hlywiak, J., Hu, W., Peng, Z., Day, D. A., Knote, C., Brune, W. H., de Gouw, J. A. and Jimenez, J. L.: Modeling the Radical Chemistry in an Oxidation Flow Reactor: Radical Formation and Recycling, Sensitivities, and the OH Exposure Estimation Equation., J. Phys. Chem. A, 119(19), 4418–4432, doi:10.1021/jp509534k, 2015.

Mazzoleni, L. R., Zielinska, B. and Moosmüller, H.: Emissions of levoglucosan, methoxy phenols, and organic acids from prescribed burns, laboratory combustion of wildland fuels, and residential wood combustion, Environ. Sci. Technol., 41(7), 2115–2122, doi:10.1021/es061702c, 2007.

McNeill, V. F., Yatavelli, R. L. N., Thornton, J. A., Stipe, C. B. and Landgrebe, O.: Heterogeneous OH oxidation of palmitic acid in single component and internally mixed aerosol particles: Vaporization and the role of particle phase, Atmos. Chem. Phys., 8(17), 5465–5476, doi:10.5194/acp-8-5465-2008, 2008.

Samburova, V., Connolly, J., Gyawali, M., Yatavelli, R. L. N., Watts, A. C., Chakrabarty, R. K., Zielinska, B., Moosmüller, H. and Khlystov, A.: Polycyclic aromatic hydrocarbons in biomass-burning emissions and their contribution to light absorption and aerosol toxicity, Sci. Total Environ., 568, 391–401, doi:10.1016/j.scitotenv.2016.06.026, 2016.

Wang, L., Atkinson, R. and Arey, J.: Dicarbonyl products of the OH radical-initiated reactions of naphthalene and the C1- and C2-alkylnaphthalenes, Environ. Sci. Technol., 41(8), 2803–2810, doi:10.1021/es0628102, 2007.

Yatavelli, R. L. N., Chen, L.-W. A., Knue, J., Samburova, V., Gyawali, M., Watts, A. C., Chakrabarty, R. K., Moosmüller, H., Hodzic, A., Wang, X., Zielinska, B., Chow, J. C. and Watson, J. G.: Emissions and Partitioning of Intermediate-Volatility and Semi-Volatile Polar Organic Compounds (I/SV-POCs) During Laboratory Combustion of Boreal and Sub-Tropical Peat, Aerosol Sci. Eng., 1(1), 25–32, doi:10.1007/s41810-017-0001-5,

2017.
* * *

---

## Author Comment (AC2) · 7 May 2020

Reviewer 2. This study quantified a number of compounds produced by burning six different fuels, most of them peat, and examined the effect of aging of the biomass burning smoke by reactions with OH (and to some extent ozone) in an oxidation flow reactor (OFR). The authors report emission factors (EFs) for all the compounds as well as the change in the effective EFs after OFR.

AC: We thank the reviewer 2 for the constructive review of our paper. First we re-

sponded to major comments and then we addressed specific comments. All corrections were highlighted in the manuscript.

Reviewer 2: Major comments 1. The reported dataset is comprehensive but it is not very clear to the reader what one can do with these data. While the decrease in the EFs can likely be modeled with known OH rate constants, the increasing EFs are hard to interpret without knowing whether additional products came from gas-ozone chemistry or from heterogeneous oxidation of particles by OH. There are some hints of this discussion in the paper, for example, on line 554. It would be good to expand this discussion.

AC: We acknowledge the reviewer's concern regarding interpretation of the reported EFs for the aged samples. For example, for some fuels (Malaysian and Malaysian agricultural peat) hexanoic acid levels (EFs) were higher after the OFR-oxidation, while for Pskov peat there was an obvious decrease after the OFR (Figure 4c, Page 33). The formation of hexanoic acid after the OFR can be explained by the oxidation of alkanes in fresh BB emissions. However, to confirm this theory, more analyses with a simplified set of compounds are needed. Considering the fact that we analyzed only 84 specific polar compounds in a very complex BB mixture, it is difficult to interpret the formation of different products without speculation. We also acknowledge that the oxidation on the filters during the sampling can affect chemistry of the collected BB aerosols. However, considering high concentration of OH radicals and residence time of BB emissions in the OFR (Bhattarai et al, 2018), we believe that OH radical reactions in the gas-phase prevails over the gas-phase ozone chemistry and heterogeneous oxidation in the particulate-phase inside the OFR chamber. This discussion is presented on Page 36 and we also added one more sentence that highlights this issue (Page 10, line 246-247)

Reviewer 2: Major comments 2. I always find it surprising how small the mass fraction of quantified compounds is despite clearly sophisticated analytical chemistry methods. Would it be possible to discuss what the rest of the compounds could be? Are there

giant peaks in the GC data that are not reported because of the lack of standards?

AC: We appreciate the reviewer's concern about quantifying only a small fraction of the total organic aerosol mass. In earlier studies, semivolatile organic compounds from biomass burning emissions were analyzed semi-quantitatively through both online (Fortenberry et al., 2018) and offline (Jen et al., 2019) sample collection techniques. Those studies also tried to extablish mass balance closure of different compound classes but only with identified fraction of organic aerosol mass. However, no attempt was made to compare the indentified compounds to the total organic mass. In our work, we performed full quantification of 84 compounds and we found those compounds account only for 5-7% of total organic aerosol mass.

Yes, the reviewer suggestion regarding other unknown species that can be present in the extracts (unidentified peaks on GCMS chromatograms) is absolutely right. We do observe some high-intensity peaks on full-scan GC-MS chromatograms of our samples. For example, NIST library search indicates that one of those peaks is ribose type sugars. Currently we are working on identification of these unknown species. Further results on this ongoing work will be presented in our forthcoming publication.

Specific comments:

1. Lines 126-133, 144-154: These sentences belong to either introduction or discussion. In the experimental section, it is better to focus on the specifics of the fuels used (such as water content, carbon content, etc.), not on the general importance of peatland ecosystems and eucalyptus forests

AC: In the Introduction section we added the paragraph on the importance of peat fuels per reviewer suggestion (Pages 3-4, lines 71-79), however we think it is still better to keep some description of fuels in the Experimental section too.

2. Line 165: it would probably help to know which molecule comes from which company.

AC: The company names were added (Page 8, lines 194-195) and highlighted in the reviewed manuscript.

3. Line 215: I do not think it is a sufficient argument. Reactions with OH are limited to a short time the mixture spends in OFR. In contrast, residual ozone comes out from OFR and has a chance to react with compounds on filter for the duration of sampling. Therefore, secondary oxidation of BB compounds on filter by ozone is quite possible and can only be ruled out if the residual ozone is low. The authors discuss these effects later in the paper on line 760.

AC: Please see our response on the major comment 1.

4. lines 61-72: This study uses peat as the main BB fuel. Be more specific about the importance of peat burning and why there needs to be a global emphasis on peat burning emissions.

AC: We agree and we emphasized the importance of analyzed fuels in the reviewed manuscript (Pages 3-4, lines 71-79).

5. Line 287: MW should be 124.14 g/mol

AC: Typo with the error removed (Page 14, line 328)

6. Line 360: missing right parenthesis

AC: Parenthesis was added (Page 17, line 399)

7. Line 492: replace )( with a semicolon

AC: The problem with the reference was fixed (Page 24, lines 529-530).

8. Lines 568, 660: you already used LMW on line 448, it would be better to define it there, or not use this abbreviation at all, as it is used only a few times.

AC: Thank you for the suggestion. We corrected the text and use "low MW" and "high MW" everywhere in the text. The "MW" abbreviation is explained on Page 1, line 29.

9. Line 664: error in 2.4+/-0.001 appears to be unrealistically small considering other errors reported in this paper.

AC: The reviewer is right. This error can not be that small and was due to a typo. This typo was corrected (Page 32, line 701).

10. Figure 4c: two panels have only one number on the X-axis, and ticks are too small making it hard to estimate the values from the figure (the tick comments applies to all similar figure).

AC: Ticks were fixed for Figures 4a, 4b, 4c (Pages 29, 31, and 33).

11. Figure 5: the text (on line 736) suggests that similar figures exists for other fuel types, however, they are not presented in the supporting information section

AC: The figures (Fig. S3 and Fig. S4) were added to the SM.

References:

Fortenberry, C.F., Walker, M.J., Zhang, Y., Mitroo, D., Brune, W.H., Williams, B.J., 2018. Bulk and molecular-level characterization of laboratory-aged biomass burning organic aerosol from oak leaf and heartwood fuels. Atmos. Chem. Phys. 18, 2199–2224. doi:10.5194/acp-18-2199-2018

Jen, C.N., Hatch, L.E., Selimovic, V., Yokelson, R.J., Weber, R., Fernandez, A.E., Kreisberg, N.M., Barsanti, K.C., Goldstein, A.H., 2019. Speciated and total emission factors of particulate organics from burning western US wildland fuels and their dependence on combustion efficiency. Atmos. Chem. Phys. 19, 1013–1026. doi:10.5194/acp-19-1013-2019